

# Quantum scrambling and state dependence
# of the butterfly velocity

**Xizhi Han ( 韩希之 ) and Sean A. Hartnoll**

Department of Physics, Stanford University, Stanford, California, USA

## Abstract

Operator growth in spatially local quantum many-body systems defines a scrambling velocity. We prove that this scrambling velocity bounds the state dependence of the out-of-time-ordered correlator in local lattice models. We verify this bound in simulations of the thermal mixed-field Ising spin chain. For scrambling operators, the butterfly velocity shows a crossover from a microscopic high temperature value to a distinct value at temperatures below the energy gap.


# 1   Introduction

Strongly quantum many-body systems have been important in condensed matter [1, 2] and nuclear physics [3, 4] for some time and are likely to become increasingly important with the ongoing development of quantum information processing technology [5–7]. It is essential to understand the spatio-temporal dynamics of these systems in highly quantum regimes where semiclassical methods such as the Boltzmann equation are inapplicable.

Significant progress has been made recently by considering quantum scrambling in many-body systems [8–13]. Quantum scrambling arises when operator growth under Heisenberg time evolution redistributes local information to non-local degrees of freedom. It has been found that scrambling in spatially local systems is characterized by both a rate and a velocity, e.g. [14–17]. These universal properties are manifested in the so-called out-of-time-ordered correlator (OTOC):

$$\mathcal{C}(\boldsymbol{x}, t; \rho) \equiv \text{tr}\left(\rho\, [O_1(0, t), O_2(\boldsymbol{x}, 0)]^\dagger [O_1(0, t), O_2(\boldsymbol{x}, 0)]\right), \tag{1}$$

defined for local operators $O_1, O_2$ in state $\rho$. The OTOC has been found to reveal a 'light cone' spread of quantum information, with two state-dependent characteristics: the quantum Lyapunov exponent $\lambda$ and the butterfly velocity $v_B$. Just outside the light cone (or 'butterfly cone') $|\boldsymbol{x}| \gtrsim v_B t$ for $t > 0$, the OTOC grows as the front is approached according to [18, 19]:

$$\mathcal{C}(\boldsymbol{x}, t; \rho) \sim e^{-\lambda(|\boldsymbol{x} - \boldsymbol{x}_0|/v_B - t)^{1+p}/t^p}. \tag{2}$$

In systems with many local degrees of freedom (e.g. large $N$ systems) the exponent $p = 0$ and the growth is exponential. This case is reminiscent of the classical butterfly effect. In spin lattice systems, generally $p > 0$, so that the front broadens as it spreads.

The butterfly velocity is a state-dependent speed of information propagation that is universally present in local systems, plausibly controlling important physical processes such as transport in strongly quantum regimes [20–26]. The state dependence means that the butterfly velocity is a more powerful probe of dynamics than the widely employed microscopic Lieb-Robinson velocity [27]. In this work we will show that this state dependence (e.g. temperature dependence) is tied to the underlying quantum scrambling of operators.

In quantum field theories that describe a nontrivial (quantum critical) continuum limit of lattice systems, the scaling of the butterfly velocity with temperature is $v_B \sim T^{1-1/z}$ in the simplest cases [16, 20]. The dynamical critical exponent $z$ describes the relative scaling of space and time. In this work we will characterize the butterfly velocity in general lattice models, away from critical points and without a large $N$ limit. We will obtain the temperature dependence of the butterfly velocity in quantum spin systems, extending previous infinite temperature results [19, 28]. The temperature dependence of scrambling in classical spin systems has been recently discussed in [29].

In a spatially local system the growth of operators determines a 'scrambling velocity' $v_S$, defined in (8) below. Our first result (9) states that the change of the velocity-dependent Lyapunov exponent — defined shortly in (6) — with temperature is bounded by the scrambling velocity. This result is rigorous for one-dimensional systems and plausibly true more generally.

We verify the bound in numerical simulations of the mixed-field Ising model, focusing on the temperature dependence of the butterfly velocity. In Fig. 2 below we see that the non-interacting transverse field model has a temperature-independent butterfly velocity whereas the velocity is temperature-dependent for the interacting mixed field models. In these curves, the butterfly velocity crosses over from a microscopic infinite-temperature value to a low-temperature value. The temperature scale of the crossover is set by the energy gap.

## 2 Three velocities from locality

It will be crucial to understand three different velocities that characterize spatially local quantum systems. Our results will tie these velocities together. The velocities emerge in any lattice $\Lambda$ of spins (or fermions) with a local Hamiltonian

$$H = \sum_{\boldsymbol{x} \in \Lambda} h_{\boldsymbol{x}} \ , \tag{3}$$

where $h_{\boldsymbol{x}}$ are operators localized near lattice site $\boldsymbol{x}$. Translation symmetry is not required.

### 2.1 Lieb-Robinson velocity

The Lieb-Robinson velocity defines an emergent 'light-cone' causality from local dynamics on a lattice [27]. It is a state-independent, microscopic velocity set by the magnitude of couplings in the Hamiltonian, and is insensitive to operator growth or lack thereof.

A convenient and powerful definition of $v_{\mathrm{LR}}$ is in terms of space-time rays. That is, consider an operator $O_2$ located along the ray $\boldsymbol{x} = v t \boldsymbol{n}$ (here $\boldsymbol{n}$ is a unit vector). At large times we can introduce a velocity-dependent exponent $\lambda(v)$ that determines the growth or decay of the norm of the commutator along the ray, $\|[O_1(0,t), O_2(v t \boldsymbol{n}, 0)]\| \sim e^{\lambda(v)t}$. Here $O(\boldsymbol{x}, t)$ denotes $O$ translated by a lattice vector $\boldsymbol{x}$ in space and a time $t$ with Heisenberg evolution, and $\|\cdot\|$ is the operator norm. The causal light cone defined by $v_{\mathrm{LR}}$ is such that for all $v > v_{\mathrm{LR}}$ the norm decays exponentially at late times, so that $\lambda(v) < 0$. Therefore we can define $v_{\mathrm{LR}}$ as the largest velocity such that the norm does not decay along a ray:

$$v_{\mathrm{LR}} \equiv \sup \left\{ v : \lim_{t \to \infty} \frac{1}{t} \ln \|[O_1(0,t), O_2(v t \boldsymbol{n}, 0)]\| \geq 0 \right\}. \tag{4}$$

We shall not keep the dependence on direction $\boldsymbol{n}$ and operators $O_1, O_2$ explicit.

For any $v > v_{\mathrm{LR}}$ there are ($v$-dependent) constants $\xi_{\mathrm{LR}}, C_{\mathrm{LR}} > 0$ such that for all $t, x > 0$,

$$\|[O_1(0,t), O_2(x \boldsymbol{n}, 0)]\| \leq C_{\mathrm{LR}} \|O_1\| \|O_2\| e^{(v t - x)/\xi_{\mathrm{LR}}}. \tag{5}$$

Intuitively, inequality (5) states that for $v > v_{\mathrm{LR}}$, the norm $\|[O_1(0,t), O_2(x \boldsymbol{n}, 0)]\|$ is exponentially small outside the ray $x = v t$, with a tail of length $\xi_{\mathrm{LR}}(v)$.

### 2.2 Butterfly velocity

The butterfly velocity is defined analogously to the Lieb-Robinson velocity, but using the OTOC instead of the operator norm of the commutator [10, 14]. It therefore depends on the quantum state $\rho$.

The 'velocity-dependent Lyapunov exponent' is defined by the late time growth or decay of the OTOC along a ray [18]:

$$\lambda(v; \rho) \equiv \lim_{t \to \infty} \frac{1}{t} \ln \mathcal{C}(v t, t; \rho). \tag{6}$$

Analogously to the Lieb-Robinson case, the butterfly velocity can now be defined as

$$v_B(\rho) \equiv \sup\{v : \lambda(v\boldsymbol{n};\rho) \geq 0\}, \tag{7}$$

which is state-dependent. The operator norm bounds the OTOC and hence $0 \leq v_B(\rho) \leq v_{\mathrm{LR}}$.

## 2.3 Scrambling velocity

The Lieb-Robinson bound (5) implies that the size of an operator can grow at most polynomially in time (as $t^d$ in a $d$-dimensional system). In contrast, the growth can be exponential without spatial locality, such as in SYK models [30–32]. Operator growth under Heisenberg evolution in quantum systems with a local Hamiltonian will therefore define another velocity. We will call this the 'scrambling velocity' $v_S$. For example, in strongly scrambling models, such as random unitary circuits [33–36], generic operators quickly grow into a superposition of product operators with radius $\sim v_{\mathrm{LR}}t$. In this case $v_S = v_{\mathrm{LR}}$.

More precisely, we define the scrambling velocity as follows. Given local operators $O_1$ and $O_2$, the commutator $[O_1(0,t), O_2(\boldsymbol{x},0)]$ will grow along the ray $\boldsymbol{x} = \boldsymbol{v}t$. We are interested in the growth of the operator itself rather than its norm or OTOC. Let $R(\boldsymbol{x},t)$ be the radius of support of the commutator[1] and define

$$v_S(\boldsymbol{v}) \equiv \lim_{t \to \infty} \frac{R(\boldsymbol{v}t,t)}{t}. \tag{8}$$

This is a velocity-dependent velocity because the growth of the operator can depend on the ray that we follow, just like the exponents in (4) and (6) above. This operator growth is illustrated in Fig. 1.

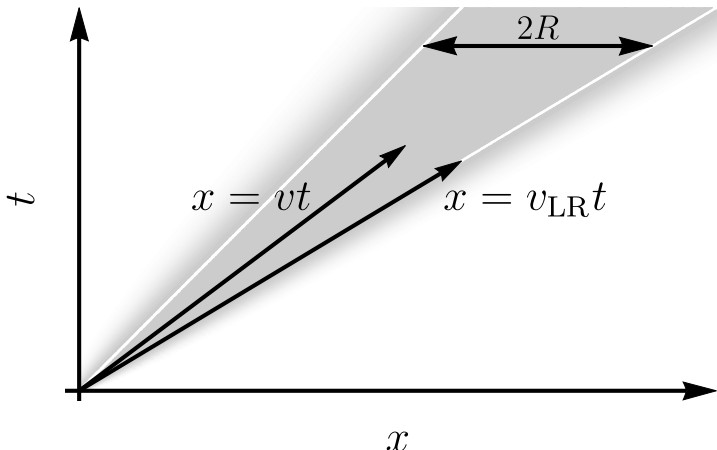

Figure 1: **Operator growth along a ray:** Schematic plot showing the definition of $R(\boldsymbol{v}t,t)$. The shaded region shows the radius of support of $O \equiv [O_1(0,t), O_2(\boldsymbol{x},0)]$ along the ray $\boldsymbol{x} = \boldsymbol{v}t$. $R$ is the radius of the support up to an exponential tail. Because of the Lieb-Robinson bound for $O_1(0,t)$ and that $O_2(\boldsymbol{x},0)$ sits on the line $\boldsymbol{x} = \boldsymbol{v}t$, the support contains the ray $\boldsymbol{x} = \boldsymbol{v}t$ and is within the Lieb-Robinson cone.

In the random circuit, let $O_1$ and $O_2$ be two single-site operators. Inside the Lieb-Robinson cone, i.e. for $|\boldsymbol{x}| \leq v_{\mathrm{LR}}t$, the commutator $[O_1(0,t), O_2(\boldsymbol{x},0)]$ has the same support as $O_1(0,t)$ so $R(\boldsymbol{x},t) = v_{\mathrm{LR}}t$ and $v_S(\boldsymbol{v}) = v_{\mathrm{LR}}$ for $|\boldsymbol{v}| \leq v_{\mathrm{LR}}$. For general systems and for $|\boldsymbol{v}| \leq v_{\mathrm{LR}}$ we

---

[1]The radius of an operator $O$ is the minimal distance $R$ such that $O$ is supported in a ball (centered at an arbitrary site) of radius $R$. Throughout the main text 'support' should be understood as up to an exponentially decaying tail. Exponential tails are discussed in detail in the appendices.

expect that $0 \leq v_S(\boldsymbol{v}) \leq v_{\mathrm{LR}}$. A proof of this statement, along with more precise definitions and technical details, is collected in the appendices.

The definition (8) also captures the absence of scrambling in non-interacting theories. A non-interacting field obeys $\phi(\boldsymbol{x}, t) = \int d\boldsymbol{y}\, f(\boldsymbol{y}, \boldsymbol{x}; t)\phi(\boldsymbol{y}, 0)$, for some function $f(\boldsymbol{y}, \boldsymbol{x}; t)$. Although the support of the operator $\phi(\boldsymbol{x}, t)$ spreads out as $t$ increases, it remains a superposition of local operators. Consider the conjugate pair $(\phi, \pi)$. It follows that $[\phi(0, t), \pi(\boldsymbol{x}, 0)] = if(\boldsymbol{x}, 0; t)$. This is a $c$-number and its support has radius $R(\boldsymbol{x}, t) = 0$. Hence $v_S(\boldsymbol{v}) = 0$ for any $\boldsymbol{v}$.

Even in non-interacting theories, however, more general operators — such as a pair of entangled quasiparticles moving in opposite directions — can have a nonzero scrambling velocity according to the definition (8). Relatedly, simple operators in weakly interacting theories need not have a small scrambling velocity. In this work we will mostly be interested in strongly scrambling systems. The bound we obtain will not, in general, usefully constrain weakly scrambling dynamics.

## 3  Scrambling bounds the state dependence of the OTOC

In the following subsections we prove a bound on the temperature dependence of the velocity-dependent Lyapunov exponent (6), in one spatial dimension. We also make an argument that an analogous result holds in higher dimensions. Namely:

$$|\partial_\beta \lambda(\boldsymbol{v}; \rho)| \leq \frac{2h}{a}\Big(v_S(\boldsymbol{v}) - (\xi + \xi_{\mathrm{LR}})\lambda(\boldsymbol{v}; \rho)\Big), \tag{9}$$

where $\beta$ is the inverse temperature, $a$ the lattice spacing, $\xi$ the correlation length, $\xi_{\mathrm{LR}}$ the microscopic lengthscale in (5), essentially the interaction range, and $h \equiv 2\sup_{\boldsymbol{x} \in \Lambda} \|h_{\boldsymbol{x}}\|$ for the Hamiltonian in (3). The content of (9) is that the change with temperature of the Lyapunov exponent along a ray is bounded by the rate of growth of the commutator along the ray. Zooming in on the butterfly light cone $v \sim v_B$, this bound implies that the growth of the commutator at the butterfly light cone bounds the change of characteristics such as the butterfly velocity. As (for example) the temperature is increased, these growing operators are 'activated' and contribute to scrambling.

A generalization, with full proof in the appendices, is as follows: For any Gibbs state $\rho = e^{-\sum_i \mu_i C^i}/\mathrm{tr}\, e^{-\sum_i \mu_i C^i}$ with mutually commuting conserved charges $C^i$, where $\mu_i \in \mathbb{R}$ and $C^i = \sum_{\boldsymbol{x} \in \Lambda} c_{\boldsymbol{x}}^i$ is a sum of local operators, then

$$\left|\frac{\partial \lambda(\boldsymbol{v}; \rho)}{\partial \mu_i}\right| \leq \frac{2c^i}{a}\Big(v_S(\boldsymbol{v}) - (\xi + \xi_{\mathrm{LR}})\lambda(\boldsymbol{v}; \rho)\Big). \tag{10}$$

The definition of $c^i > 0$ is similar to $h$ above: $c^i \equiv 2\sup_{\boldsymbol{x} \in \Lambda} \|c_{\boldsymbol{x}}^i\|$.

### 3.1  Outline of proof in one dimension

The following gives an outline of the proof of (9). The logic is straightforward, but technical complications arise, for example, due to the fact that time evolution generates exponentially decaying tails in space for local operators, so one cannot assume that local operators have strictly finite support. These technical points are addressed in the appendices.

Let $\rho = e^{-\beta H}/\mathrm{tr}\, e^{-\beta H}$ be a thermal state with inverse temperature $\beta$ and correlation length $\xi$. The steps will be as follows: *(i)* Differentiate the OTOC with respect to the inverse temperature, *(ii)* show that the main contribution to this derivative is from operators inside the support of the commutator, and *(iii)* balance the growth of this contribution, due to the growing size

of the commutator along a ray, with the growth or decay of the OTOC. We now outline these steps.

(i) *Temperature derivative of the OTOC.* Taking the derivative of the OTOC (1) with respect to the inverse temperature gives

$$\partial_\beta \mathcal{C}(\boldsymbol{x}, t; \rho) = -\text{tr}(\rho\, \widetilde{H} O^\dagger O) = -\text{tr}(\widetilde{H} \sqrt{\rho} O^\dagger O \sqrt{\rho}), \tag{11}$$

where $O \equiv i[O_1(0, t), O_2(\boldsymbol{x}, 0)]$ and $\widetilde{H} \equiv H - \text{tr}(\rho H)$ is the Hamiltonian with thermal expectation value subtracted out.

The Hamiltonian $H$ in (3) is written as a sum of local terms. We can split this sum up into terms that are inside and outside the support of the commutator $O$ (for some location $\boldsymbol{x}$ and time $t$). As in the definition of $v_S$, let $O$ be roughly supported in a ball of center $\boldsymbol{y}_0$ and radius $R$. Then

$$\widetilde{H} = \sum_{|\boldsymbol{y}-\boldsymbol{y}_0|\leq R+\delta} \widetilde{h}_{\boldsymbol{y}} + \sum_{|\boldsymbol{y}-\boldsymbol{y}_0|>R+\delta} \widetilde{h}_{\boldsymbol{y}}\,, \tag{12}$$

where $\delta > 0$ can take any value. As for $\widetilde{H}$, $\widetilde{h}_{\boldsymbol{y}} \equiv h_{\boldsymbol{y}} - \text{tr}(\rho h_{\boldsymbol{y}})$. This decomposition can now be inserted into the derivative (11).

(ii) *Dominance by operators inside the commutator.* We first bound the contribution from outside of the support of the commutator, with $|\boldsymbol{y} - \boldsymbol{y}_0| > R + \delta$ in (12). Due to the thermal correlation length $\xi$, the connected correlation function of $\widetilde{h}_{\boldsymbol{y}}$ with $O^\dagger O$ will decay exponentially in the distance $|\boldsymbol{y} - \boldsymbol{y}_0|$. Thus, for some constant $C > 0$ and all $\boldsymbol{y} \in \Lambda$ such that $|\boldsymbol{y} - \boldsymbol{y}_0| > R$: $|\text{tr}(\widetilde{h}_{\boldsymbol{y}} \sqrt{\rho} O^\dagger O \sqrt{\rho})| \leq C\|\widetilde{h}_{\boldsymbol{y}}\|\|O\|^2 e^{(R-|\boldsymbol{y}-\boldsymbol{y}_0|)/\xi}$. Summing over $|\boldsymbol{y} - \boldsymbol{y}_0| > R + \delta$, the contribution to (11) from operators outside of the commutator is bounded by

$$\sum_{|\boldsymbol{y}-\boldsymbol{y}_0|>R+\delta} \left|\text{tr}(\widetilde{h}_{\boldsymbol{y}} \sqrt{\rho} O^\dagger O \sqrt{\rho})\right| \leq C' \sup_{\boldsymbol{y}\in\Lambda}\|\widetilde{h}_{\boldsymbol{y}}\|\|O\|^2 e^{-\delta/\xi}\,. \tag{13}$$

In $d$ spatial dimensions and for $R + \delta \gg \xi$, $C' \sim C\xi(R + \delta)^{d-1}/a^d$ from doing the sum over $|\boldsymbol{y} - \boldsymbol{y}_0| > R + \delta$ ($a$ is the lattice spacing). There is a technical subtlety in obtaining (13) due to the need to commute factors of $\sqrt{\rho}$ through $\widetilde{h}_{\boldsymbol{y}}$; we deal with this in the appendices.

We can similarly bound the contribution to (11) from operators inside the support of the commutator, with $|\boldsymbol{y}-\boldsymbol{y}_0| \leq R+\delta$. As in the main text, define the maximal local coupling in the Hamiltonian as

$$h \equiv 2 \sup_{\boldsymbol{y}\in\Lambda}\|h_{\boldsymbol{y}}\|\,. \tag{14}$$

Note that $\|\widetilde{h}_{\boldsymbol{y}}\| \leq 2\|h_{\boldsymbol{y}}\|$, so that

$$|\text{tr}(\widetilde{h}_{\boldsymbol{y}} \sqrt{\rho} O^\dagger O \sqrt{\rho})| \leq \|\widetilde{h}_{\boldsymbol{y}}\| \text{tr}(\rho\, O^\dagger O) \leq h\, \mathcal{C}(\boldsymbol{x}, t; \rho)\,. \tag{15}$$

Notice that the inequality still goes through if we take

$$h = \sup_{\boldsymbol{y}\in\Lambda} \frac{|\text{tr}(\widetilde{h}_{\boldsymbol{y}} \sqrt{\rho} O^\dagger O \sqrt{\rho})|}{\text{tr}(\rho\, O^\dagger O)}\,. \tag{16}$$

Now, the number of terms in the first sum of (12) is $V_{R+\delta}$, the number of lattice points in a ball of radius $R + \delta$. Therefore, putting together (13) and (15), we can bound the derivative (11) by:

$$|\partial_\beta \mathcal{C}(\boldsymbol{x}, t; \rho)| \leq V_{R+\delta}\, h\, \mathcal{C}(\boldsymbol{x}, t; \rho) + C' h\, \|O\|^2 e^{-\delta/\xi}\,. \tag{17}$$

We will see that in a certain kinematic limit, the final term in (17), from outside of the support of the commutator, is small compared to the other terms.

(iii) *Bounding the derivative by the growth of the commutator*: The inequality (17) simplifies at late times along a ray $\boldsymbol{x} = \boldsymbol{v}t$. From the definition (6) of the velocity-dependent Lyapunov exponent, $\mathcal{C}(\boldsymbol{v}t, t; \rho) \sim e^{\lambda(\boldsymbol{v};\rho)t}$ as $t \to \infty$. We furthermore set $\delta = (-\xi\lambda(\boldsymbol{v};\rho) + \epsilon)t > 0$, with $\epsilon > 0$ a small number. This choice is such that the final term in (17) decays exponentially faster than the others as $t \to \infty$. This final term is therefore negligible in this limit. In this way, as $t \to \infty$ the following inequality is obtained:

$$|\partial_\beta \lambda(\boldsymbol{v};\rho)| \le h \lim_{t \to \infty} \frac{V_{R-\xi\lambda(\boldsymbol{v};\rho)t}}{t} . \tag{18}$$

This expression bounds the temperature dependence of the Lyapunov exponent in terms of the late time growth of the commutator along a ray. The late time limit in (18) is manifestly finite in one spatial dimension, $d = 1$. In one dimension at large radii $V_r \approx 2r/a$, where $a$ is the lattice spacing. In this case, the operator growth in (18) is precisely given by the scrambling velocity defined in (8). Thus, in terms of the scrambling velocity we obtain (A more rigorous treatment in the appendices, allowing for exponential tails in the support, shows that $\xi \to \xi + \xi_{\mathrm{LR}}$. We include this shift in the following statement of the bound.)

$$|\partial_\beta \lambda(\boldsymbol{v};\rho)| \le \frac{2h}{a}\Big(v_S(\boldsymbol{v}) - (\xi + \xi_{\mathrm{LR}})\lambda(\boldsymbol{v};\rho)\Big). \tag{19}$$

## 3.2 Generalization to higher dimensions

In higher dimensions, $V_r$ will scale as $r^d$ for $d > 1$ and hence the late time bound (18) is always trivially true. However, we conjecture that the bound stated in (9) holds for arbitrary dimensions, based on a Lieb-Robinson type argument. One way of understanding the Lieb-Robinson bound is to expand

$$O_1(t) = \sum_{n=0}^{\infty} \frac{(it[H, \cdot])^n}{n!} O_1 = O_1 + it[H, O_1] - \frac{t^2}{2}[H, [H, O_1]] + \dots , \tag{20}$$

and observe that in the expansion, for $[O_1(0, t), O_2(\boldsymbol{x}, 0)]$ to be nonzero, a commutator sequence of local terms in $H$ connecting $O_1$ and $O_2$ is necessary, which starts at order $n \approx |\boldsymbol{x}|/R_H$ where $R_H$ is the range of local terms in $H$. For such a high order term to be significant, $t$ has to be later than $|\boldsymbol{x}|/(R_H h)$ and this gives an estimate of $v_{\mathrm{LR}} \approx R_H h$.

In a proof along these lines it is intuitively clear that outside the Lieb-Robinson cone $|\boldsymbol{x}| = v_{\mathrm{LR}}t$, the leading contributions to the commutator $[O_1(0, t), O_2(\boldsymbol{x}, 0)]$ come from $O_1$ taking commutators with local terms in $H$ (as shown in (20)), via the shortest path from the origin to $\boldsymbol{x}$. Hence it is plausible that the operator $[O_1(0, t), O_2(\boldsymbol{x}, 0)]$, for $|\boldsymbol{x}| \gg v_{\mathrm{LR}}t$, is approximately one-dimensional, along the line connecting 0 and $\boldsymbol{x}$. Then the bound (9) is still expected to be true, although possibly with a larger 'renormalized' $h$.

# 4 Temperature dependence of the butterfly velocity

## 4.1 Numerical results on the mixed field Ising chain

To motivate the general discussion of butterfly velocities, it will be useful to have some explicit numerical results for the temperature dependence of the butterfly velocity at hand. To this end

we have studied the mixed field Ising chain with Hamiltonian

$$H = -J \sum_{i=1}^{N-1} Z_i Z_{i+1} + h_X \sum_{i=1}^{N} X_i + h_Z \sum_{i=1}^{N} Z_i \,, \tag{21}$$

where $X_i$, $Y_i$ and $Z_i$ are Pauli matrices at site $i$. Numerics is done with a straightforward generalization of the Matrix Product Operator (MPO) method discussed in [19, 28] to finite temperatures. Some analytic results on OTOCs in the transverse field model ($h_Z = 0$) can be found in [37]. In numerics we will have $N = 25$. More details can be found in the appendices. Results for the temperature dependence of the butterfly velocity for Pauli $Z$ operators are shown in Fig. 2.

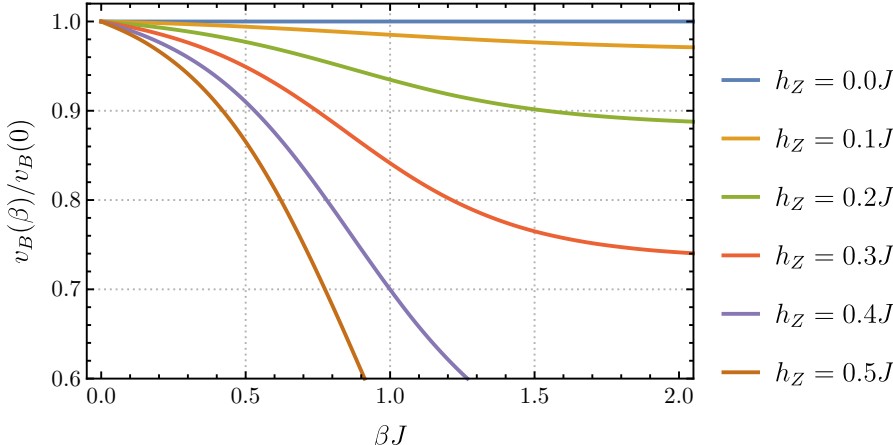

Figure 2: **Temperature-dependent butterfly velocity in the mixed field Ising chain** (21) with $h_X = 1.05J$ and different $h_Z$. The inverse temperature is denoted as $\beta$. The model with $h_Z = 0$ is dual to free fermions and has a temperature-independent butterfly velocity. The appendices contain more details about numerics and error estimates.

The numerical results in Fig. 2 exhibit the behavior advertised in the introduction, and which we will understand in detail below. The transverse field Ising model ($h_Z = 0$) is dual to free fermions via a Jordan-Wigner transformation. The longitudinal field $h_Z$ introduces interactions. We expect interactions to induce scrambling dynamics and hence a nontrivial temperature dependence of the butterfly velocity, and this is what the figure shows.

The temperature-independent butterfly velocity of the transverse field model deserves some elaboration. There are two points to make. Firstly, the transverse field model is special in its duality to a *non-interacting* integrable system, where $v_S = 0$ for the commutator of fermion creation and annihilation operators, for example. For interacting integrable systems, typically $v_S > 0$ and the butterfly velocity is state-dependent [38]. Indeed, we have verified numerically that the butterfly velocity is temperature-dependent in such models. Interacting integrable systems are scrambling, even while they are not chaotic.

Secondly, in the transverse field model, Pauli $Z$'s in the spin frame are dual to nonlocal fermion chains by the Jordan-Wigner transformation. Due to this nonlocality, our inequality doesn't apply in the fermion frame. In fact, even local operators describing small numbers of quasiparticles in a non-interacting theory can have $v_S > 0$ by our definition because entangled pairs of quasiparticles moving in opposite directions technically lead to a linearly growing radius of support for the operator. We believe that it may be possible to overcome this technical complication in the future with an improved definition of the scrambling velocity, such that $v_S = 0$ for spatially separated but entangled non-scrambling operators. Indeed, we shall now

argue that the butterfly velocity is temperature independent for all local operators in a non-interacting system.

In a non-interacting theory the propagation of quasiparticles is independent of the state they are propagating in, due to the absence of interactions between them. While the quasiparticles may have a nontrivial dispersion and hence temperature-dependent average velocity, any local operator includes modes of all wavevectors and, in particular, maximal velocity modes. Thus we expect $v_B$ is independent of the state. Therefore, the temperature-independence of the butterfly velocity observed in our numerics is indeed symptomatic of the non-interacting integrability of the system.

### 4.2 Bounding the butterfly velocity

The temperature dependence shown in Fig. 2 can be understood from the connections between the OTOC and scrambling velocity that we have described. The 'light front' form (2) for the OTOC implies that the velocity-dependent Lyapunov exponent is

$$\lambda(v; \rho) = -\lambda(v/v_B - 1)^{1+p} \qquad \text{for} \qquad v \geq v_B. \tag{22}$$

This precise form for $\lambda(v; \rho)$ is conveniently explicit, but the only qualitatively essential aspect for our results is the presence of a 'butterfly cone'. As we explained above, in general $\lambda$, $v_B$ and $p \geq 0$ are state-dependent. Therefore, the $\partial_{\mu_i}$ derivative in (10) will act on each of these quantities. Substituting the specific form (22) for $\lambda(v; \rho)$ into (10), for $v > v_B$, leads to the following slightly complicated expression:

$$a\lambda(\Delta v)^{1+p}\left|\partial_{\mu_i}\ln\lambda + \ln(\Delta v)\partial_{\mu_i}p - (1+p)\frac{v/v_B}{\Delta v}\partial_{\mu_i}\ln v_B\right|$$
$$\leq 2c^i\left[v_S(v) + (\xi + \xi_{\text{LR}})\lambda(\Delta v)^{1+p}\right], \tag{23}$$

where $\Delta v \equiv v/v_B - 1 > 0$ is a dimensionless measure of how far the velocity is outside the butterfly cone. A simple consequence of (23) follows, when there is no scrambling. Suppose that $v_S(v) = 0$. In that case, taking $\Delta v \to 0^+$, the leading term on the left side of (23) is the last one. It follows that

$$v_S = 0 \implies \partial_{\mu_i} v_B = 0. \tag{24}$$

Hence $v_B$ is constant for operators that do not scramble. We noted above, however, that this result is not directly applicable to the transverse field Ising chain.

Increasing variation of $v_B$ with temperature is observed in Fig. 2 as integrability is increasingly broken by turning on $h_Z$ in the mixed field Ising model. The crossover temperature in Fig. 2 is set by the energy gap $\Delta$ (of order $J$ for $h_Z = 0.1 \sim 0.5J$), as we now explain. Intuitively, one might expect $v_B$ to cease varying at temperatures $T \ll \Delta$. This is what is seen in the numerical data. We can argue for this by improving an aspect of the proof outlined previously. As we note there, the proof still goes through if we take $h$ in (9) to be instead given by

$$h = \sup_{t>0, y\in\Lambda} \frac{|\text{tr}(\tilde{h}_y\sqrt{\rho}\,O^\dagger O\sqrt{\rho})|}{\text{tr}(\rho\,O^\dagger O)}, \tag{25}$$

where $O \equiv i[O_1(0, t), O_2(\mathbf{v}t, 0)]$ and $\tilde{h}_y \equiv h_y - \text{tr}(\rho h_y)$. This is not an especially tractable expression in general, but it can be evaluated for a gapped system at zero temperature, where $\rho \equiv |0\rangle\langle 0|$. In that case $h = \sup_{y\in\Lambda}\langle 0|\tilde{h}_y|0\rangle = 0$, where now $\tilde{h}_y \equiv h_y - \langle 0|h_y|0\rangle$. Hence in gapped systems at low temperatures, we may set $h \approx 0$ in the bound (9). It follows that $\partial_\beta v_B \to 0$ when $T \to 0$ in a gapped system, consistent with the finite low temperature butterfly velocities seen in Fig. 2.

The numerical results in Fig. 3 substantiate the above argument, suggesting that $\partial_\beta v_B$ decays exponentially as $\beta\Delta \to \infty$. In Fig. 3 the bound has furthermore been written as a bound on the derivative of the butterfly velocity, and is found to be most constraining at intermediate temperatures and with strong scrambling, where it is within an order of magnitude of the true value.

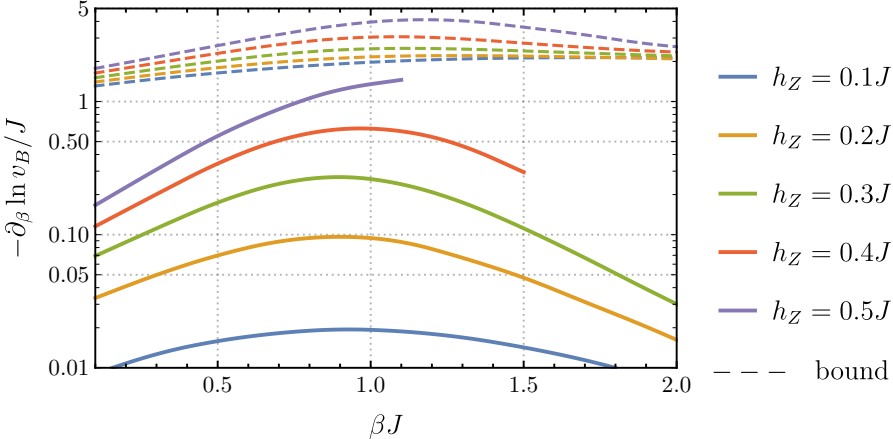

Figure 3: **Bounding the temperature derivative of the butterfly velocity:** Temperature derivative of the butterfly velocity in mixed field Ising chains, with $h_X = 1.05J$ and different $h_Z$ in (21). The inverse temperature is denoted as $\beta$. The bound (23) is shown as the dashed curves. In the bound $v_S$ is replaced by $3Ja$ ($a = 1$ is the lattice spacing), using the fact that $v_S \leq v$ for $v = 3Ja$ and $\xi_{\mathrm{LR}} = a$ in the Lieb-Robinson inequality (5), in the spin duality frame. Curves are cut off when estimated error is significant (see the appendices for more details).

Our bound combined together with numerics leads to a consistent picture of the temperature dependence of the butterfly velocity in chaotic spin systems with a gap $\Delta$. Stronger scrambling allows for stronger temperature dependence of $v_B$, which furthermore approaches a constant at $T \ll \Delta$. These facts explain the crossover features of the curves in Fig. 2. More quantitatively, the overall variation $v_B(\beta = 0)/v_B(\beta = \infty)$ can be bounded by integrating our bound from $\beta = 0$ to $\beta\Delta \sim 1$ (assuming that there are no intervening thermal phase transitions). For small $v_S(v)$, this integration can be done explicitly, leading to a bound on the change in the butterfly velocity from infinite to zero temperature. For notational convenience let $v_S^B \equiv v_S(v_B)$. At small $v_S^B$ one may take $\Delta v \sim (v_S^B/v_B)^{1/(1+p)}$ in (23) and the leading term on the left hand side is again the final one, which integrates to

$$\left| \ln \frac{v_B(\beta = \infty)}{v_B(\beta = 0)} \right| \lesssim \int_0^{1/\Delta} d\beta \, \frac{2h v_B^{p/(1+p)}[1 + (\xi + \xi_{\mathrm{LR}})\lambda/v_B]}{a\lambda(1+p)} \left( v_S^B \right)^{1/(1+p)} , \qquad (26)$$

to leading order in $v_S^B \to 0$. Typically $v_B(\beta = 0) \sim v_{\mathrm{LR}}$. Schematically we can therefore write

$$v_B(T = 0) \gtrsim v_{\mathrm{LR}} \, e^{-\alpha v_S^\gamma/\Delta} . \qquad (27)$$

Here $\alpha$ is a dimensionful constant, $\gamma$ a dimensionless constant and we have singled out the $v_S$ and $\Delta$ dependences. It follows that *(i)* as $v_S \to 0$, $\ln v_B$ can vary as a power $v_S^\gamma$ of the scrambling velocity, and *(ii)* if the gap $\Delta \to 0$, $v_B$ may approach zero at $T = 0$. Indeed, power law butterfly velocities $v_B \sim T^{1-1/z}$, with $z$ the dynamical critical exponent, are found in strongly chaotic gapless holographic models [16, 20].

## 5  Final comments

In summary, we have shown how locality of quantum dynamics ties operator growth to the butterfly velocity. This connection arises because the growth of the spatial support of the commutator right outside the butterfly cone bounds the change of the butterfly velocity with e.g. temperature. The butterfly velocity is state-dependent and therefore gives a richer characterization of the finite temperature dynamics than is possible from the microscopic Lieb-Robinson velocity alone. We have demonstrated these ideas explicitly in numerical studies of quantum chaotic lattice models at finite temperature. Looking forward, we hope that the methods we have developed can be used to bound other important quantities that underpin quantum many-body systems, in particular the thermalization length and time, as well as transport observables such as the thermal diffusivity.

## Acknowledgements

It is a pleasure to acknowledge Jacob Marks for helping with numerics and Daren Chen for reading the proofs. We are grateful to Vedika Khemani and Xiao-Liang Qi for insightful comments on an earlier version. SAH is partially funded by DOE award de-sc0018134. XH is supported by a Stanford Graduate Fellowship. Computational work was performed on the Sherlock cluster at Stanford University, with the ITensor library for implementing tensor network calculations.

## Appendices

This appendix contains six sections: section A sets up notations and backgrounds for discussions that follow. In section B we review the Lieb-Robinson, Araki and correlation length bounds used in our proof. Precise definitions for Lieb-Robinson, butterfly and scrambling velocities are given in section C and we prove several inequalities regarding them. Section D collects technical lemmas for exponentially local operators and section E gives a rigorous proof of the general results. Details of numerical implementations and data analysis are presented in section F.

## A  Notation

In this section we introduce notations and concepts necessary for a rigorous proof of our result. The bound will be formulated for a lattice[2] $\Lambda$ of spins in $d$ spatial dimensions, and rigorously proved for $d = 1$. There are isomorphic finite-dimensional Hilbert spaces $\mathcal{H}_x$ associated to each lattice site $x \in \Lambda$ and denote $\mathcal{B}_x$ as the space of linear operators acting on $\mathcal{H}_x$. An operator $O$ is said to be supported on a subset $S \subset \Lambda$ if $O \in \bigotimes_{x \notin S} \mathbb{C}I \otimes \bigotimes_{x \in S} \mathcal{B}_x$, i.e. $O$ is a sum of product operators that are identity outside $S$. The minimal set that $O$ is supported on is called the support of $O$, denoted as $\operatorname{supp} O$.[3]

To better characterize the spatial distribution of operators, define superoperators $\mathcal{P}_S$ and $\mathcal{Q}_S \equiv \operatorname{Id} - \mathcal{P}_S$ such that $\mathcal{P}_S$ is the projection onto the subspace $\bigotimes_{x \notin S} \mathbb{C}I \otimes \bigotimes_{x \in S} \mathcal{B}_x$. That is, $\mathcal{P}_S$ projects onto operators supported on $S$ (so $\mathcal{P}_S[O] = O$ if $O$ is supported on $S$). More explicitly

---

[2]Technically the infinite lattice should be thought as the limit of a sequence of increasing finite subsystems. We will not delve into subtleties related to this point.

[3]Note $\operatorname{supp} O = \emptyset$ if and only if $O = cI$ for some $c \in \mathbb{C}$.

$$\mathcal{P}_S[O] \equiv \int_{\text{supp}\,U \cap S = \emptyset} dU \, U O U^\dagger, \tag{28}$$

where the integral is Haar averaging over unitaries outside $S$. However, note $\mathcal{Q}_S$ is *not* the projection onto operators supported on $\Lambda - S$. Consider an example of two sites $\Lambda = \{1,2\}$ and an operator $O = O_1 \otimes O_2$, where neither $O_1$ nor $O_2$ is the identity. By definition, $0 = \mathcal{P}_1[O] = \mathcal{P}_2[O] \neq \mathcal{Q}_1[O] = \mathcal{Q}_2[O] = O$.

Henceforth if the subscript $S = \{x\}$ is a single-element set, $\mathcal{P}_{\{x\}}$ and $\mathcal{Q}_{\{x\}}$ are written as $\mathcal{P}_x$ and $\mathcal{Q}_x$ for short. Also define the superoperator $\mathcal{P}_T^r$ with a superscript $r > 0$ as $\mathcal{P}_S$ for $S = \{y \in \Lambda : \exists\, x \in T, |x - y| < r\}$, i.e. projection onto operators supported within a distance $r$ from the set $T$, and $\mathcal{Q}_T^r \equiv \text{Id} - \mathcal{P}_T^r$.

From (28) we have the following inequalities:

$$\|\mathcal{P}_S[O]\| \leq \|O\|, \quad \|\mathcal{Q}_S[O]\| = \|O - \mathcal{P}_S[O]\| \leq \|O\| + \|\mathcal{P}_S[O]\| \leq 2\|O\|, \tag{29}$$

as $\|U\| = \|U^\dagger\| = 1$. Also $\mathcal{P}_S[I] = I$, $\mathcal{Q}_S[I] = 0$ for any $S \subset \Lambda$. Unless otherwise specified, $\|O\|$ will always denote the operator norm, i.e. the maximal singular value of $O$.

We will be interested primarily in operators that are "exponentially local", denoted as $\mathcal{B}(x, R; \xi, C)$. We say $O \in \mathcal{B}(x, R; \xi, C)$ with $x \in \Lambda$, $R, C \geq 0$ and $\xi > 0$, if for any $r \geq R$,

$$\|\mathcal{Q}_x^r[O]\| \leq C \|O\| e^{-(r-R)/\xi}. \tag{30}$$

Intuitively, this means $O$ is supported on the ball of radius $R$ and centered at $x$, up to an exponential tail of lengthscale $\xi$. Operators supported on a finite number of sites (called "finitely supported") are of course exponentially local as well. We shall assume the Hamiltonian is a sum of finitely supported hermitian terms:

$$H = \sum_\alpha J_\alpha H^\alpha, \quad H^\alpha \equiv \sum_{x \in \Lambda} h_x^\alpha, \quad h_x^\alpha \in \mathcal{B}(x, R_H; 0^+, 0), \tag{31}$$

which also defines $R_H > 0$ and $\alpha$ labels different couplings in the Hamiltonian. Translational invariance is not necessary but $\|h^\alpha\| \equiv \sup_{x \in \Lambda} \|h_x^\alpha\|$ should be bounded.

A Gibbs state is a density matrix of the form

$$\rho = e^{-\sum_i \mu_i C^i} / \text{tr}\, e^{-\sum_i \mu_i C^i}, \tag{32}$$

for some $\mu_i \in \mathbb{R}$ and

$$C^i \equiv \sum_{x \in \Lambda} c_x^i, \quad c_x^i \in \mathcal{B}(x, R_H; 0^+, 0). \tag{33}$$

In the proof it is *not* required that $[C^i, C^j] = 0$. With only one $i$, with $\mu$ the inverse temperature and with $C = H$, $\rho$ is the thermal density matrix.

# B  Review of locality bounds

In this section we review some established locality bounds. First is the Lieb-Robinson bound in local lattice systems [27, 39–41]. This both bounds the spread of support of a local operator by the distance $v|t|$, where $t$ is the real time of Heisenberg evolution, and also implies an emergent causality with $v$ acting as the "speed of light". For a discussion of the relation between (i) and (ii) in the following theorem, see section 3 of [42].

**Theorem 1** (Lieb-Robinson)**.** *There exist $v, \xi_{\text{LR}}, C_{\text{LR}} > 0$, dependent on lattice geometry and Hamiltonian, such that*

*(i) for any $t \in \mathbb{R}$, $r > 0$ and operator $O$,*

$$\|\mathcal{Q}^r_{\text{supp}\,O}[O(t)]\| \leq C_{\text{LR}}|\partial \,\text{supp}\,O|\|O\| \min\{1, e^{(v|t|-r)/\xi_{\text{LR}}}\} \,, \tag{34}$$

*where $|\partial S|$ is the number of lattice links (say, between $x$ and $y$) such that $x \in S$ but $y \notin S$;*

*(ii) for any $t \in \mathbb{R}$, operators $O_1$ and $O_2$,*

$$\|[O_1(t), O_2]\| \leq C_{\text{LR}} \min\{|\partial \,\text{supp}\,O_1|, |\partial \,\text{supp}\,O_2|\}\|O_1\|\|O_2\| \min\{1, e^{(v|t|-d)/\xi_{\text{LR}}}\} \,, \tag{35}$$

*where $d = \min\{|x - y| : x \in \text{supp}\,O_1, y \in \text{supp}\,O_2\}$ is the distance between the support of $O_1$ and $O_2$.*

In this bound $v \sim \sum_\alpha |J_\alpha| \|h^\alpha\| R_H$, recall (31), i.e. coupling times range of local terms in the Hamiltonian, and $\xi_{\text{LR}} \sim R_H$. So quantities in the Lieb-Robinson bound are set by microscopic scales, to be differentiated from the butterfly velocity, which is an analog of a "renormalized" Lieb-Robinson velocity in thermal states [16].

Next is the Araki bound [42–44] extending the Lieb-Robinson bound to *complex* times. Note the theorem is specific to one dimension [44] and $l_A(\mu_i)$ may be exponential in $|\mu_i|$; in this sense the restriction is weaker for complex time evolution:

**Theorem 2** (Araki)**.** *In one dimension, for any Gibbs state $\rho$ as defined in (32) but with $\mu_i \in \mathbb{C}$, there exist $l_A(\mu_i), C_A(\mu_i), \xi_A > 0$, dependent on lattice geometry and charges $C^i$, such that for any finitely supported operator $O$ and $r \geq l_A(\mu_i)$,*

$$\|\rho O \rho^{-1}\| \leq C_A(\mu_i)|\text{supp}\,O|\|O\|, \tag{36}$$

$$\|\mathcal{Q}^r_{\text{supp}\,O}[\rho O \rho^{-1}]\| \leq C_A(\mu_i)|\text{supp}\,O|\|O\|e^{(l_A(\mu_i)-r)/\xi_A} \,, \tag{37}$$

*where $|\text{supp}\,O|$ is the number of sites in $\text{supp}\,O$.*

Note, however, from the proof of the Araki bound (e.g., Theorem 3.1 of [44]) one can see that there are Araki inequalities as stated in Theorem 2 for *arbitrarily* small $\xi_A$, at the expense of a possibly large $l_A$. Later in the proof of our bound only $\xi_A$ enters the final expression; hence at that time one can take $\xi_A \to 0$ as a large $l_A$ doesn't affect the result.

Originally the Araki bound is only stated for finitely supported operators but it is straightforward to generalize it to exponentially local ones. Such generalization will be useful in proving our bound, so a proof is given in section D.

Finally we would like to introduce some exponential clustering theorems: for particular kinds of states, equal-time connected correlations decay exponentially in space. More precisely for a state (density matrix) $\rho$, the correlation length of $\rho$ is the $\xi > 0$ that is optimal with respect to the following property: there exists $C > 0$ and a function $l_0(\cdot) > 0$ such that for any operators $O_1$ and $O_2$ (supported on sets $S$ and $T$) sufficiently far apart, i.e., $d \geq l_0(\delta)$,

$$|\text{tr}(\rho \, O_1 O_2) - \text{tr}(\rho \, O_1)\text{tr}(\rho \, O_2)| \leq C\delta\|O_1\|\|O_2\|e^{-d/\xi}, \tag{38}$$

where $\delta \equiv \min\{|\partial S|, |\partial T|\}$ is the number of lattice links crossing the boundary of $S$ or $T$, and $d \equiv \min\{|x - y| : x \in S, y \in T\}$ is the distance between two sets. Note that in this statement, $O_1$ and $O_2$ could be *any*, not necessarily local, operators.

Existence of a finite $\xi > 0$ with the property stated around (38) has been proved for (i) one-dimensional Gibbs states [43] (restricted to local operators $O_1$ and $O_2$), (ii) $\rho = |0\rangle\langle 0|$ where $|0\rangle$ is the unique ground state of a gapped Hamiltonian [40, 45], and (iii) thermal states $\rho \propto \exp(-\beta H)$ in general dimensions at sufficiently high temperatures [46] (clearly $\xi \to 0$ when $\beta \to 0$). Of course the Hamiltonians associated with these states must be local, as in (31) above. It is plausible that the correlation length $\xi$ as defined around (38) is finite for Gibbs states $\rho$ in general systems with local dynamics and away from phase transitions.

## C  Definitions of velocities

In this section we define precisely the (possibly anisotropic) Lieb-Robinson, butterfly and scrambling velocities introduced in the main text and prove the bound $v_B, v_S \leq v_{LR}$. For definiteness fix a class of local operators, denoted as $\mathcal{O}$; for example, $\mathcal{O}$ could be all single-site operators with unit norm, localized at origin. The Lieb-Robinson bound Theorem 1 (ii) can be stated for such operators along any particular direction $\boldsymbol{n}$:

**Theorem 3** (Operator-dependent anisotropic Lieb-Robinson)**.** *For any direction $\boldsymbol{n}$ and operator $O_1, O_2 \in \mathcal{O}$, there exist $v$, $\xi_{LR}$, $C_{LR} > 0$, dependent on $\boldsymbol{n}$, $O_1$, $O_2$, lattice geometry and Hamiltonian, such that for any $t > 0$, $x > 0$,*

$$\| [O_1(0, t), O_2(x\boldsymbol{n}, 0)] \| \leq C_{LR} \|O_1\| \|O_2\| \min\{1, e^{(vt-x)/\xi_{LR}}\} . \tag{39}$$

From Theorem 3 one immediate candidate for defining the Lieb-Robinson velocity is

$$v_{LR}^{(1)}(\boldsymbol{n}; O_1, O_2) \equiv \inf\{v > 0 : \exists \xi_{LR}, C_{LR} > 0 \text{ with the property stated in Theorem 3}\} , \tag{40}$$

that is, the smallest velocity with a Lieb-Robinson inequality. However such a definition shows some disadvantages in numerical or experimental applications: it is inaccurate to fit data to exponential tails because the theorem only states an inequality (not an equality), and in fact in many lattice systems of interest the tail is observed to be sub-exponential (e.g., Gaussian) [18, 19]; also it is impractical, if not impossible, to decide whether such $\xi_{LR}$ and $C_{LR}$ exist for all times, from only a finite number of data points.

A more convenient definition is found in the original Lieb-Robinson paper [27]

$$v_{LR}^{(2)}(\boldsymbol{n}; O_1, O_2) \equiv \sup \left\{ v : \lim_{t \to \infty} \frac{1}{t} \ln \| [O_1(0, t), O_2(vt\boldsymbol{n}, 0)] \| \geq 0 \right\} . \tag{41}$$

We will assume that the limit exists and is a continuous function of $v$. By definition $v_{LR}^{(2)}$ gives a causality "lightcone" outside which (for $x/t > v$) the commutator vanishes exponentially at late times.

It is relatively easy to see that $v_{LR}^{(1)} \geq v_{LR}^{(2)}$:

**Proposition 1.** *For any direction $\boldsymbol{n}$ and operators $O_1, O_2 \in \mathcal{O}$, we have $v_{LR}^{(1)}(\boldsymbol{n}; O_1, O_2) \geq v_{LR}^{(2)}(\boldsymbol{n}; O_1, O_2)$.*

*Proof.* Let $v > 0$ belong to the set in (40), i.e., there exist $\xi, C > 0$ such that for all $x, t > 0$, $\| [O_1(0, t), O_2(x\boldsymbol{n}, 0)] \| \leq C\|O_1\| \|O_2\| \min\{1, e^{(vt-x)/\xi}\}$. Then, for any $v' > v$, $\lim_{t \to \infty} t^{-1} \ln \| [O_1(0, t), O_2(v't\boldsymbol{n}, 0)] \| \leq \lim_{t \to \infty} t^{-1} \ln(C\|O_1\| \|O_2\| e^{(v-v')t/\xi}) = (v - v')/\xi < 0$, and hence any $v' > v$ is not contained in the set in (41). Therefore the supremum $v_{LR}^{(2)}$ is at most $v$. This is true for any $v > 0$ in the set of (40), hence $v_{LR}^{(2)} \leq v_{LR}^{(1)}$. $\square$

Conversely to show that $v_{LR}^{(1)} \leq v_{LR}^{(2)}$, we need the following lemma:

**Lemma 1.** *For any positive functions $f(x, t)$ and $g(x, t)$, if limits*

$$\lim_{t \to \infty} \frac{1}{t} \ln f(vt, t) = \lambda_f(v), \quad \lim_{t \to \infty} \frac{1}{t} \ln g(vt, t) = \lambda_g(v) , \tag{42}$$

*exist, are uniform for $v \in [v_0, \infty)$, and $\lambda_f(v) + a < \lambda_g(v)$ for some $a > 0$ and all $v \geq v_0$, then there is $t_0 > 0$ that*

$$f(x, t) < g(x, t) \quad \forall x \geq v_0 t, \, t \geq t_0 . \tag{43}$$

*Proof.* Because the limits (42) are uniform, for any $\varepsilon > 0$ there is $T(\varepsilon) > 0$ such that for any $t \geq T(\varepsilon)$ and $v \geq v_0$, $\ln f(vt, t)/t < \lambda_f(v) + \varepsilon$, $\ln g(vt, t)/t > \lambda_g(v) - \varepsilon$. Now choose $\varepsilon = a/2$ and $t_0 = T(a/2)$, we have $\ln f(vt, t)/t < \lambda_f(v) + a/2 < \lambda_g(v) - a/2 < \ln g(vt, t)/t$ hence $f(vt, t) < g(vt, t)$, for all $t \geq t_0$, $v \geq v_0$. $\qquad\square$

**Proposition 2.** $v_{\mathrm{LR}}^{(1)}(\mathbf{n}; O_1, O_2) \leq v_{\mathrm{LR}}^{(2)}(\mathbf{n}; O_1, O_2)$, *given the limit in (41) is uniform for all* $v > v_{\mathrm{LR}}^{(2)}(\mathbf{n}; O_1, O_2)$.

*Proof.* We would like to prove the proposition in the following two steps:

Step one: For any $v > v_{\mathrm{LR}}^{(2)}$, we show that (i) implies (ii), and (ii) implies (iii), where

(i) $\lim_{t \to \infty} t^{-1} \ln \|[O_1(0, t), O_2(v'tn, 0)]\| < 0$ for any $v' \geq v$;

(ii) $\exists \varepsilon, \xi > 0$ that $\lim_{t \to \infty} t^{-1} \ln \|[O_1(0, t), O_2(v'tn, 0)]\| \leq (v - v')/\xi - \varepsilon$ for any $v' \geq v$;

(iii) $\exists C, \xi > 0$ that $\|[O_1(0, t), O_2(x\mathbf{n}, 0)]\| \leq C\|O_1\|\|O_2\| \min\{1, e^{(vt-x)/\xi}\}$ for $x, t > 0$.

Step two: By definition (41) we have for any $v > v_{\mathrm{LR}}^{(2)}$, (i) holds for $v$; so (iii) is true for $v$ as well, and $v$ should be in the set on the right-hand side of (40) hence $v_{\mathrm{LR}}^{(1)} \leq v$. This shows that $v_{\mathrm{LR}}^{(1)} \leq v_{\mathrm{LR}}^{(2)}$.

So now it remains to prove that (i) $\Rightarrow$ (ii) and (ii) $\Rightarrow$ (iii):

(i) $\Rightarrow$ (ii): For clarity let's denote $\lambda(v) \equiv \lim_{t \to \infty} t^{-1} \ln \|[O_1(0, t), O_2(vt\mathbf{n}, 0)]\|$, then (i) says that $\lambda(v') < 0$ for any $v' \geq v$ and to arrive at (ii) we hope to find $\varepsilon, \xi > 0$ such that $\lambda(v') \leq (v - v')/\xi - \varepsilon$ for all $v' \geq v$.

Before construction of $\varepsilon$ and $\xi$, it is remarkable that there is a restriction on $\lambda(v')$ from Theorem 3: the Lieb-Robinson bound states that there are some $C_0, v_0, \xi_0 > 0$ such that $\lambda(v') \leq \lim_{t \to \infty} t^{-1} \ln(C_0\|O_1\|\|O_2\|e^{(v_0 - v')t/\xi_0}) = (v_0 - v')/\xi_0$ for all $v' > 0$.

We shall construct $\varepsilon > 0$ first. Note $(v - v')/\xi \leq 0$ for $v' \geq v$, hence it is required that $\lambda(v') \leq -\varepsilon$ for all $v' \geq v$. So we may choose $\varepsilon = \inf_{v' \geq v}(-\lambda(v')/2) \geq 0$. To show that $\varepsilon > 0$, we have to check that $-\lambda(v') > 0$ is bounded from zero on $[v, \infty)$. The only concern is $\lambda(v')$ may be arbitrarily close to zero when $v' \to \infty$; but this is not possible because from the previous paragraph $-\lambda(v') \geq (v' - v_0)/\xi_0 \to \infty$ as $v' \to \infty$. Hence $\varepsilon > 0$ is well-defined in this way.

Then to satisfy $\lambda(v') \leq (v - v')/\xi - \varepsilon$ for all $v' \geq v$, choose ($\xi_0$ is there for future convenience) $\xi \equiv \max\{\xi_0, \sup_{v' \geq v}(v - v')/(\lambda(v') + \varepsilon)\}$ (as constructed in the last paragraph the denominator is always negative). The task is then to show that $\xi < \infty$; similarly the only place things could go wrong is when $v' \to \infty$, but in that limit $|\lambda(v') + \varepsilon| \geq |\lambda(v')|/2 \geq (v' - v_0)/2\xi_0$ hence $\lim_{v' \to \infty}(v - v')/(\lambda(v') + \varepsilon) \leq 2\xi_0$ is bounded. So $\xi > 0$ is well-defined as well and (ii) is proved.

(ii) $\Rightarrow$ (iii): We would like to apply the Lemma 1 for $f(x, t) = \|[O_1(0, t), O_2(x\mathbf{n}, 0)]\|$ and $g(x, t) = \|O_1\|\|O_2\|e^{(vt-x)/\xi}$. Note in this case $\lambda_f(v') = \lambda(v') \leq (v - v')/\xi - \varepsilon = \lambda_g(v') - \varepsilon$ for any $v' \geq v$. Then by the lemma there is $t_0 > 0$ such that $\|[O_1(0, t), O_2(x\mathbf{n}, 0)]\| \leq \|O_1\|\|O_2\|e^{(vt-x)/\xi}$ for all $x \geq vt$ and $t \geq t_0$. Hence for (iii) to hold it suffices to choose that $C \equiv \max\{2, \sup_{0 < x < vt \text{ or } 0 < t < t_0} f(x, t)/g(x, t)\}$. As before we have to check that the supremum is not infinite. We will discuss the three cases (a) $0 < x < vt$, (b) $0 < t < t_0$ with $x \geq v_0 t$, and (c) $0 < t < t_0$ with $0 < x < v_0 t$ separately.

For $0 < x < vt$, $f(x, t)/g(x, t) = \|[O_1(0, t), O_2(x\mathbf{n}, 0)]\|/\|O_1\|\|O_2\|e^{(vt-x)/\xi}$ is less than $\|[O_1(0, t), O_2(x\mathbf{n}, 0)]\|/\|O_1\|\|O_2\| \leq 2$. So indeed $f(x, t)/g(x, t)$ is bounded in this region.

For $0 < t < t_0$ and $x \geq v_0 t$, $f(x, t)/g(x, t) = \|[O_1(0, t), O_2(x\mathbf{n}, 0)]\|/\|O_1\|\|O_2\|e^{(vt-x)/\xi}$ can be bounded using the Lieb-Robinson Theorem 3: there is some $C_0, v_0, \xi_0 > 0$ such that $\|[O_1(0, t), O_2(x\mathbf{n}, 0)]\| \leq C_0\|O_1\|\|O_2\|e^{(v_0 t - x)/\xi_0} \leq C_0\|O_1\|\|O_2\|e^{(v_0 t - x)/\xi}$ (by construction $\xi \geq \xi_0$) so $f(x, t)/g(x, t) \leq C_0 e^{(v_0 - v)t/\xi}$ which is a bounded function for $0 < t < t_0$.

Finally for $0 < t < t_0$ and $0 < x < v_0 t$, $f(x,t)/g(x,t)$ is bounded because it is continuous and the region is bounded. Hence we've shown that $C > 0$ is well-defined and with $\xi$ appearing in (ii), (iii) is true. $\qquad\square$

Henceforth the Lieb-Robinson velocity will be defined as $v_{\text{LR}} \equiv v_{\text{LR}}^{(1)} = v_{\text{LR}}^{(2)}$. The technical uniformity condition is true for known examples. The same proof shows the equivalence of two definitions of the butterfly velocity. For future use only the definition corresponding to $v_{\text{LR}}^{(2)}$ is recorded:

$$v_B(\boldsymbol{n}; O_1, O_2, \rho) \equiv \sup\left\{v : \lim_{t\to\infty} \frac{1}{t} \ln \mathcal{C}_{O_1 O_2}(v t \boldsymbol{n}, t; \rho) \geq 0\right\}, \tag{44}$$

where the OTOC $\mathcal{C}_{O_1 O_2}(\boldsymbol{x}, t; \rho)$ is defined in (1). As the velocity-dependent quantum Lyapunov exponent is defined as in (6), an equivalent definition of $v_B$ reads:

$$v_B(\boldsymbol{n}; O_1, O_2, \rho) \equiv \sup\{v : \lambda_{O_1 O_2}(v \boldsymbol{n}; \rho) \geq 0\}. \tag{45}$$

As expected, the butterfly velocity in any state is bounded by the Lieb-Robinson velocity:

**Proposition 3.** $v_B(\boldsymbol{n}; O_1, O_2, \rho) \leq v_{\text{LR}}(\boldsymbol{n}; O_1, O_2)$ *for any* $O_1, O_2 \in \mathcal{O}$*, density matrix* $\rho$ *and direction* $\boldsymbol{n}$*.*

*Proof.* This follows from definition (41) and (44), and $\mathcal{C}_{O_1 O_2}(\boldsymbol{x}, t; \rho) \leq \|[O_1(0, t), O_2(\boldsymbol{x}, 0)]\|^2$. $\qquad\square$

Finally the scrambling velocity can be precisely defined in the language of exponentially local operators, defined around (30). Let $O \equiv i[O_1(0, t), O_2(\boldsymbol{v} t, 0)]$, then[4]

$$v_S(\boldsymbol{v}; O_1, O_2, \xi) \equiv \inf_{C > 0} \overline{\lim_{t\to\infty}} \frac{1}{t} \inf\{R \geq 0 : \exists \boldsymbol{x} \in \Lambda, O \in \mathcal{B}(\boldsymbol{x}, R; \xi, C)\}, \tag{46}$$

where the smallest ball, with radius $R$ and centered at $\boldsymbol{x}$, is understood as roughly the "support" of the commutator $O$. The quantities $\xi$ and $C$ characterize the exponential tail that we neglected in the main text. Clearly $v_S \geq 0$ and decreases with increasing $\xi$.

For any triple $(\boldsymbol{v}, \xi \equiv \xi_{\text{LR}}, C_{\text{LR}})$ from Theorem 1, we now show that $v_S(\boldsymbol{v}; \xi) \leq v$. Thus we have an upper bound of $v_S$ by velocities with a Lieb-Robinson inequality. Note the $\xi$-dependence of $v_S$ was omitted in the main text. More precisely, if $O_2(\boldsymbol{v} t, 0)$ is within the "support" of $O_1(0, t)$, for scrambling systems at late times we would expect $\|[O_1(0, t), O_2(\boldsymbol{v} t, 0)]\|$ to equilibrate to a nonzero constant value; if so, $v_S \leq v$:

**Proposition 4.** *Given* $\boldsymbol{v}, \xi > 0, O_1, O_2 \in \mathcal{O}$*, if for any* $t > 0, O_1(0, t) \in \mathcal{B}(0, vt; \xi, C)$ *for some* $v > |\boldsymbol{v}|, C > 0$ *and* $\underline{\lim}_{t\to\infty} \|[O_1(0, t), O_2(\boldsymbol{v} t, 0)]\| > 0$*, then* $v_S(\boldsymbol{v}; O_1, O_2, \xi) \leq v$*.*

*Proof.* Let $O(t) \equiv [O_1(0, t), O_2(\boldsymbol{v} t, 0)]$, $c \equiv \underline{\lim}_{t\to\infty} \|O(t)\| > 0$. As $|\boldsymbol{v}| < v$, $\mathcal{Q}_0^r[O_2(\boldsymbol{v} t, 0)] = 0$ for $r \geq vt$ at late times. Then $O(t) = [\mathcal{P}_0^r[O_1(0, t)], \mathcal{P}_0^r[O_2(\boldsymbol{v} t, 0)]] + [\mathcal{Q}_0^r[O_1(0, t)], \mathcal{P}_0^r[O_2(\boldsymbol{v} t, 0)]]$. But the first term is supported in the ball of radius $r$ centered at origin, so $\|\mathcal{Q}_0^r[O(t)]\| = \|\mathcal{Q}_0^r[\mathcal{Q}_0^r[O_1(0, t)], \mathcal{P}_0^r[O_2(\boldsymbol{v} t, 0)]]\| \leq 4\|\mathcal{Q}_0^r[O_1(0, t)]\|\|\mathcal{P}_0^r[O_2(\boldsymbol{v} t, 0)]\| \leq 4C\|O_1\|\|O_2\|e^{(vt-r)/\xi}$, where we have used the definition (30) that for all $t > 0$ and $r \geq vt$, $\|\mathcal{Q}_0^r[O_1(0, t)]\| \leq C\|O_1\|e^{(vt-r)/\xi}$ with the inequalities (29).

So there is a time $t_0 > 0$ that for all $t > t_0$, $\|O(t)\| \geq c/2$ as well as $\|\mathcal{Q}_0^r[O(t)]\| \leq 4C\|O_1\|\|O_2\|e^{(vt-r)/\xi}$ for all $r \geq vt$. Hence $\|\mathcal{Q}_0^r[O(t)]\| \leq C'\|O(t)\|e^{(vt-r)/\xi}$, for all $t > t_0$ and $r \geq vt$, if we choose $C' = 8C\|O_1\|\|O_2\|/c$. That is, $O(t) \in \mathcal{B}(0, vt; \xi, C')$ for $t > t_0$ hence by definition (46), $v_S(\boldsymbol{v}; O_1, O_2, \xi) \leq v$. $\qquad\square$

All velocities can be maximized over direction $\boldsymbol{n}$ to recover their isotropic definitions, or over $O_1, O_2 \in \mathcal{O}$ to remove the operator dependence.

---

[4]To make sure the limit exists, we have used the limit superior $\overline{\lim}$ and the limit inferior $\underline{\lim}$.

# D  Bounds for exponentially local operators

In this section we collect some lemmas and generalize Theorem 2 and the exponential clustering condition (38) to exponentially local operators. Readers are encouraged to review sections A and B. The following inequality will be useful: for any $A, B \geq 0$ and $k, \gamma > 0$,

$$\sum_{n=\lceil k \rceil}^{\infty} (An + B)e^{-\gamma n} \leq (Ak + A + B)e^{-\gamma k}(1 - e^{-\gamma})^{-2} , \tag{47}$$

where $\lceil x \rceil$ denotes the least integer greater than or equal to $x$. To show this, by doing the summation exactly it is easy to check that for any $A, B \geq 0$, $\gamma > 0$ and integer $m \geq 1$,

$$\sum_{n=m}^{\infty} (An + B)e^{-\gamma n} \leq (Am + B)e^{-\gamma m}(1 - e^{-\gamma})^{-2} , \tag{48}$$

and the inequality (47) follows because if $m = \lceil k \rceil$, $m \leq k + 1$ in the linear factor and $k \leq m$ implies that $e^{-\gamma m} \leq e^{-\gamma k}$ as well.

The following lemma bounds the product of two exponentially local operators:

**Lemma 2.** *Let $O_1 \in \mathcal{B}(\boldsymbol{x}, R; \xi_1, C_1)$ and $O_2 \in \mathcal{B}(\boldsymbol{x}, R; \xi_2, C_2)$, then for any $r \geq R$,*

$$\|\mathcal{Q}_{\boldsymbol{x}}^r[O_1 O_2]\| \leq 2(C_1 + C_2)\|O_1\|\|O_2\|e^{(R-r)/\max\{\xi_1, \xi_2\}} . \tag{49}$$

*Proof.* Note that for any $r > 0$, $O_1 O_2 = \mathcal{P}_{\boldsymbol{x}}^r[O_1]\mathcal{P}_{\boldsymbol{x}}^r[O_2] + O_1 \mathcal{Q}_{\boldsymbol{x}}^r[O_2] + \mathcal{Q}_{\boldsymbol{x}}^r[O_1]\mathcal{P}_{\boldsymbol{x}}^r[O_2]$, and $\mathcal{Q}_{\boldsymbol{x}}^r[\mathcal{P}_{\boldsymbol{x}}^r[O_1]\mathcal{P}_{\boldsymbol{x}}^r[O_2]] = 0$. So by (29) and (30), for $r \geq R$,

$$\begin{aligned}
\|\mathcal{Q}_{\boldsymbol{x}}^r[O_1 O_2]\| &\leq 2\|O_1\|\|\mathcal{Q}_{\boldsymbol{x}}^r[O_2]\| + 2\|\mathcal{Q}_{\boldsymbol{x}}^r[O_1]\|\|O_2\| \\
&\leq 2C_2\|O_1\|\|O_2\|e^{(R-r)/\xi_2} + 2C_1\|O_1\|\|O_2\|e^{(R-r)/\xi_1} .
\end{aligned} \tag{50}$$

$\square$

Next is the Araki bound (cf. Theorem 2) for exponentially local operators:

**Theorem 4.** *For any one-dimensional Gibbs state $\rho$ as defined in (32) with $\mu_i \in \mathbb{C}$ and operator $O \in \mathcal{B}(\boldsymbol{x}, R; \xi, C)$, there exists $C'(\mu_i, \xi, C) > 0$ (dependent on lattice geometry and $C^i$ as well) such that for all $r \geq R + l_A(\mu_i) + a$,*

$$\|\rho O \rho^{-1}\| \leq C'(\mu_i, \xi, C)\|O\|(1 + 2R/a), \tag{51}$$

$$\|\mathcal{Q}_{\boldsymbol{x}}^r[\rho O \rho^{-1}]\| \leq C'(\mu_i, \xi, C)\|O\|[1 + 2(r - l_A(\mu_i))/a]e^{(R + l_A(\mu_i) + a - r)/(\xi_A + \xi)} . \tag{52}$$

*Here $l_A(\mu_i)$ and $\xi_A$ are those appearing in the Araki bound, and $a$ is the lattice spacing.*

*Proof.* For the first inequality, let $m \equiv \lceil (R + a)/a \rceil$. Decompose $O = \mathcal{P}_{\boldsymbol{x}}^{(m-1)a}[O] + \sum_{n \geq m} O_n$, where $O_n \equiv \mathcal{P}_{\boldsymbol{x}}^{na} \mathcal{Q}_{\boldsymbol{x}}^{(n-1)a}[O] = \mathcal{P}_{\boldsymbol{x}}^{na}[O] - \mathcal{P}_{\boldsymbol{x}}^{(n-1)a}[O]$. Then by Theorem 2 with (29) and (30), for $n \geq m$,

$$\begin{aligned}
\|\rho O_n \rho^{-1}\| = \|\rho \mathcal{P}_{\boldsymbol{x}}^{na} \mathcal{Q}_{\boldsymbol{x}}^{(n-1)a}[O]\rho^{-1}\| &\leq C_A(\mu_i)(2n+1)\|\mathcal{P}_{\boldsymbol{x}}^{na} \mathcal{Q}_{\boldsymbol{x}}^{(n-1)a}[O]\| \\
&\leq C_A(\mu_i)(2n+1)\|\mathcal{Q}_{\boldsymbol{x}}^{(n-1)a}[O]\| \leq C_A(\mu_i)(2n+1)C\|O\|e^{(R-na+a)/\xi} .
\end{aligned} \tag{53}$$

Also by Theorem 2, $\|\mathcal{P}[O]\| \leq \|O\|$ and $m \leq (R + a)/a + 1 = R/a + 2$,

$$\|\rho \mathcal{P}_{\boldsymbol{x}}^{(m-1)a}[O]\rho^{-1}\| \leq C_A(\mu_i)(2m-1)\|\mathcal{P}_{\boldsymbol{x}}^{(m-1)a}[O]\| \leq C_A(\mu_i)(2R/a + 3)\|O\| . \tag{54}$$

Sum (53) with (47) (where $A = 2$, $B = 1$, $k = (R + a)/a$ and $\gamma = a/\xi$) to get the bound

$$\|\rho O \rho^{-1}\| \leq C_{\mathrm{A}}(\mu_i)(2R/a + 3)\|O\| + C_{\mathrm{A}}(\mu_i)C(2R/a + 5)\|O\|(1 - e^{-a/\xi})^{-2} . \tag{55}$$

Denote $C_1(\mu_i, \xi, C) = 3C_{\mathrm{A}}(\mu_i) + 5C_{\mathrm{A}}(\mu_i)C(1 - e^{-a/\xi})^{-2}$, so that

$$\|\rho O \rho^{-1}\| \leq C_1(\mu_i, \xi, C)\|O\|(1 + 2R/a) . \tag{56}$$

For the second inequality, expand $O = \mathcal{P}_{\emptyset}[O] + \sum_{n \geq 0} O_n$, where $\mathcal{P}_{\emptyset}[O]$ is proportional to identity and $O_n \equiv \mathcal{P}_{\boldsymbol{x}}^{na} \mathcal{Q}_{\boldsymbol{x}}^{(n-1)a}[O] = \mathcal{P}_{\boldsymbol{x}}^{na}[O] - \mathcal{P}_{\boldsymbol{x}}^{(n-1)a}[O]$. Because $\mathcal{Q}_{\boldsymbol{x}}^r[I] = 0$,

$$\mathcal{Q}_{\boldsymbol{x}}^r[\rho O \rho^{-1}] = \sum_{n=0}^{\infty} \mathcal{Q}_{\boldsymbol{x}}^r[\rho O_n \rho^{-1}] . \tag{57}$$

Let $\delta \equiv \alpha(r - l_{\mathrm{A}}(\mu_i) - R - a) \geq 0$ for any $0 < \alpha < 1$ and split the sum (57) into two parts: $0 \leq na < R + \delta + a$ and $na \geq R + \delta + a$. Apply Theorem 2 for the first part (also note $\|O_n\| \leq 2\|O\|$ by (29)):

$$\|\mathcal{Q}_{\boldsymbol{x}}^r[\rho O_n \rho^{-1}]\| \leq 2C_{\mathrm{A}}(\mu_i)(2n + 1)\|O\|e^{(l_{\mathrm{A}}(\mu_i)+na-r)/\xi_{\mathrm{A}}} , \tag{58}$$

and further with inequalities (29) and definition (30) for the second part:

$$\|\mathcal{Q}_{\boldsymbol{x}}^r[\rho O_n \rho^{-1}]\| \leq 2\|\rho O_n \rho^{-1}\| \leq 2C_{\mathrm{A}}(\mu_i)(2n + 1)\|O_n\|$$
$$\leq 2C_{\mathrm{A}}(\mu_i)(2n + 1)\|\mathcal{Q}_{\boldsymbol{x}}^{(n-1)a}[O]\| \leq 2CC_{\mathrm{A}}(\mu_i)(2n + 1)\|O\|e^{(R-na+a)/\xi} . \tag{59}$$

Overall, sum (58) as geometric series after applying $n \leq k$ and sum (59) with (47) (where $A = 2$, $B = 1$, $k = (R + \delta + a)/a$ and $\gamma = a/\xi$):

$$\|\mathcal{Q}_{\boldsymbol{x}}^r[\rho O \rho^{-1}]\| \leq 2C_{\mathrm{A}}(\mu_i)(2k + 1)\|O\|e^{(l_{\mathrm{A}}(\mu_i)+ka-r)/\xi_{\mathrm{A}}}(1 - e^{-a/\xi_{\mathrm{A}}})^{-1}$$
$$+ 2CC_{\mathrm{A}}(\mu_i)(2k + 3)\|O\|e^{(R-ka+a)/\xi}(1 - e^{-a/\xi})^{-2}$$
$$\leq 2C_{\mathrm{A}}(\mu_i)[1 + 2(r - l_{\mathrm{A}}(\mu_i))/a]\|O\|e^{-(1-\alpha)\delta/\alpha\xi_{\mathrm{A}}}(1 - e^{-a/\xi_{\mathrm{A}}})^{-1}$$
$$+ 2CC_{\mathrm{A}}(\mu_i)[3 + 2(r - l_{\mathrm{A}}(\mu_i))/a]\|O\|e^{-\delta/\xi}(1 - e^{-a/\xi})^{-2} , \tag{60}$$

where in the second inequality we have replaced $ka = R + \delta + a$ in the exponents and applied the bound $k \leq (r - l_{\mathrm{A}}(\mu_i))/a$ (because $\alpha \leq 1$) in the prefactors. Now

$$\|\mathcal{Q}_{\boldsymbol{x}}^r[\rho O \rho^{-1}]\| \leq C_2(\mu_i, \xi, C)\|O\|[1 + 2(r - l_{\mathrm{A}}(\mu_i))/a]e^{(R+l_{\mathrm{A}}(\mu_i)+a-r)/(\xi_{\mathrm{A}}+\xi)} , \tag{61}$$

if one chooses $\alpha = \xi/(\xi_{\mathrm{A}} + \xi)$ to equate the exponents and $C_2(\mu_i, \xi, C) = 2C_{\mathrm{A}}(\mu_i)(1 - e^{-a/\xi_{\mathrm{A}}})^{-1} + 6CC_{\mathrm{A}}(\mu_i)(1 - e^{-a/\xi})^{-2}$.

Finally it suffices to choose $C'(\mu_i, \xi, C) \equiv \max\{C_1(\mu_i, \xi, C), C_2(\mu_i, \xi, C)\}$. $\qquad \square$

Observe that the operator $\rho O \rho^{-1}$ as stated in (52), is not exponentially local explicitly (due to the prefactor that is linear in $r$). To work around this the following corollary of Theorem 4 is particularly useful:

**Corollary 1.** *For any $\varepsilon > 0$, there is a $\widetilde{C}'(\mu_i, \xi, C, \varepsilon)$ such that*

$$\rho O \rho^{-1} \in \mathcal{B}\left(\boldsymbol{x}, R + l_{\mathrm{A}}(\mu_i) + a; \xi_{\mathrm{A}} + \xi + \varepsilon, \widetilde{C}'e^{\varepsilon R/(\xi_{\mathrm{A}}+\xi)^2}\|O\|/\|\rho O \rho^{-1}\|\right) . \tag{62}$$

*Proof.* First note that for $\zeta(\xi) \equiv \xi_{\mathrm{A}} + \xi$,

$$e^{R/\zeta} = e^{R/(\zeta+\varepsilon)}e^{\varepsilon R/\zeta(\zeta+\varepsilon)} \leq e^{R/(\zeta+\varepsilon)}e^{\varepsilon R/\zeta^2} , \tag{63}$$

so it suffices to find $\widetilde{C}'(\mu_i, \xi, C, \varepsilon)$ such that for all $x \equiv r - l_{\mathrm{A}}(\mu_i) - a \geq 0$,

$$C'(\mu_i, \xi, C)[1 + 2(x + a)/a]e^{-x/\zeta(\xi)} \leq \widetilde{C}'e^{-x/(\zeta(\xi)+\varepsilon)} , \tag{64}$$

which clearly exists. $\qquad \square$

Finally we generalize inequality (38) to exponentially local operators as well; for future use we will work in one dimension only:

**Theorem 5.** *Let $\rho$ be a one-dimensional state with $\xi$, $C$ and $l_0(\cdot) > 0$ as stated around (38). If $O_1 \in \mathcal{B}(\boldsymbol{x}, R_1; \xi_1, C_1)$, $O_2 \in \mathcal{B}(\boldsymbol{y}, R_2; \xi_2, C_2)$ and $|\boldsymbol{x} - \boldsymbol{y}| \geq l_0(2) + R_1 + R_2$,*

$$|\mathrm{tr}(\rho \, O_1 O_2) - \mathrm{tr}(\rho \, O_1)\mathrm{tr}(\rho \, O_2)|$$
$$\leq 2(C + C_1 + C_2)\|O_1\|\|O_2\|e^{(R_1 + R_2 + l_0(2) - |\boldsymbol{x} - \boldsymbol{y}|)/(\xi + \xi_1 + \xi_2)} \,. \tag{65}$$

*Proof.* Let $\Delta \equiv |\boldsymbol{x} - \boldsymbol{y}| - l_0(2) - R_1 - R_2 \geq 0$, and define $r \equiv R_1 + \alpha_1 \Delta$ and $s \equiv R_2 + \alpha_2 \Delta$ for $\alpha_1, \alpha_2 > 0$ and $\alpha_1 + \alpha_2 < 1$. Denote $c(O_1, O_2) \equiv \mathrm{tr}(\rho \, O_1 O_2) - \mathrm{tr}(\rho \, O_1)\mathrm{tr}(\rho \, O_2)$ for convenience and observe $|c(O_1, O_2)| \leq 2\|O_1\|\|O_2\|$. Then

$$c(O_1, O_2) = c(\mathcal{P}_{\boldsymbol{x}}^r[O_1], \mathcal{P}_{\boldsymbol{y}}^s[O_2]) + c(\mathcal{Q}_{\boldsymbol{x}}^r[O_1], \mathcal{P}_{\boldsymbol{y}}^s[O_2]) + c(O_1, \mathcal{Q}_{\boldsymbol{y}}^s[O_2]) \,. \tag{66}$$

By inequality (38), (note $\delta = 2$ if $S$ and $T$ are intervals in (38) and $\|\mathcal{P}[O]\| \leq \|O\|$)

$$|c(\mathcal{P}_{\boldsymbol{x}}^r[O_1], \mathcal{P}_{\boldsymbol{y}}^s[O_2])| \leq 2C\|O_1\|\|O_2\|e^{-l_0(2)/\xi}e^{-(1 - \alpha_1 - \alpha_2)\Delta/\xi} \,, \tag{67}$$

and by definition (30),

$$|c(\mathcal{Q}_{\boldsymbol{x}}^r[O_1], \mathcal{P}_{\boldsymbol{y}}^s[O_2])| \leq 2\|\mathcal{Q}_{\boldsymbol{x}}^r[O_1]\|\|O_2\| \leq 2C_1\|O_1\|\|O_2\|e^{-\alpha_1 \Delta/\xi_1} \,, \tag{68}$$

$$|c(O_1, \mathcal{Q}_{\boldsymbol{y}}^s[O_2])| \leq 2\|O_1\|\|\mathcal{Q}_{\boldsymbol{y}}^s[O_2]\| \leq 2C_2\|O_1\|\|O_2\|e^{-\alpha_2 \Delta/\xi_2} \,. \tag{69}$$

Now choose $\alpha_1 = \xi_1/(\xi + \xi_1 + \xi_2)$ and $\alpha_2 = \xi_2/(\xi + \xi_1 + \xi_2)$ so that the exponents with $\Delta$ are all equal. Sum them up to get (65). $\qquad\square$

# E   Proof of the bound

In this section we give a proof of the bounds stated in the main text. To avoid clutter of notations, all quantities in this section may depend on lattice geometry, Hamiltonian $H$ (31) and charges $C^i$ (33) implicitly.

**Theorem 6.** *For any one-dimensional Gibbs state $\rho$ as defined in (32) with correlation length $\xi_{\mathrm{cor}}$ (read around (38) for a definition), $\varepsilon, \delta > 0$, any operators $O_1, O_2$ and $\boldsymbol{x} \in \Lambda$, $t > 0$, there exist $A(\mu_i, \xi, C, \varepsilon)$, $B(\mu_i) > 0$ such that*

$$\left|\frac{\partial \mathcal{C}_{O_1 O_2}(\boldsymbol{x}, t; \rho)}{\partial \mu_i}\right| \leq A \sup_{\boldsymbol{y} \in \Lambda}\|c_{\boldsymbol{y}}^i\|\|O_1\|^2\|O_2\|^2(1 + 2R/a)e^{\varepsilon R/(\xi + \xi_A)^2}e^{-\delta/(\xi_{\mathrm{cor}} + \xi_A + \xi + \varepsilon)}$$
$$+ 2c^i(R + \delta + B)\mathcal{C}_{O_1 O_2}(\boldsymbol{x}, t; \rho)/a \,, \tag{70}$$

*and*

$$\left|\frac{\partial \mathcal{C}_{O_1 O_2}(\boldsymbol{x}, t; \rho)}{\partial J_\alpha}\right| \leq A\beta \sup_{\boldsymbol{y} \in \Lambda}\|h_{\boldsymbol{y}}^\alpha\|\|O_1\|^2\|O_2\|^2(1 + 2R/a)e^{\varepsilon R/(\xi + \xi_A)^2}e^{-\delta/(\xi_{\mathrm{cor}} + \xi_A + \xi + \varepsilon)}$$
$$+ 2\beta h^\alpha(R + \delta + B)\mathcal{C}_{O_1 O_2}(\boldsymbol{x}, t; \rho)/a + 2\int_0^t ds \, \sqrt{\mathcal{C}_{O_1 O_2}(\boldsymbol{x}, t; \rho)\mathcal{C}_{[H^\alpha(-s), O_1]O_2}(\boldsymbol{x}, t; \rho)} \,, \tag{71}$$

*where $a$ is the lattice spacing and $\xi_A$ is defined in Theorem 2. The inverse temperature is denoted as $\beta$ and $J_\alpha$ labels couplings in the Hamiltonian (31). Denote $O \equiv i[O_1(0, t), O_2(\boldsymbol{x}, 0)]$; $R$, $\xi$ and $C$ are such that $O \in \mathcal{B}(\boldsymbol{y}_0, R; \xi, C)$ for some $\boldsymbol{y}_0 \in \Lambda$. Finally*

$$c^i \equiv \int_0^1 ds \, c^i(s) \equiv \int_0^1 ds \, \sup_{\boldsymbol{y} \in \Lambda}\frac{|\mathrm{tr}(\rho^s \tilde{c}_{\boldsymbol{y}}^i \rho^{1-s} O^\dagger O)|}{\mathrm{tr}(\rho \, O^\dagger O)} \,, \tag{72}$$

where $\widetilde{c}_y^i \equiv c_y^i - \mathrm{tr}(\rho\, c_y^i)$, and same for $h^\alpha$ with $c_y^i$ replaced by $h_y^\alpha$. And if $C^i$ commute with each other, $c^i$ can be chosen as

$$c^i \equiv \sup_{y \in \Lambda} \frac{|\mathrm{tr}(\sqrt{\rho}\,\widetilde{c}_y^i\,\sqrt{\rho}O^\dagger O)|}{\mathrm{tr}(\rho\, O^\dagger O)} \le 2 \sup_{y \in \Lambda} \|c_y^i\|\,. \tag{73}$$

*Proof.* We start with proving (70). By definition (1) and (32),

$$\frac{\partial \mathcal{C}_{O_1 O_2}(\boldsymbol{x},t;\rho)}{\partial \mu_i} = -\int_0^1 ds\, \mathrm{tr}(\rho^s \widetilde{C}^i \rho^{1-s} O^\dagger O)\,, \tag{74}$$

where for any operator $C$, $\widetilde{C} \equiv C - \mathrm{tr}(\rho\, C)$. Now recall $C^i$ is a sum of local terms (33):

$$C^i = \sum_{y \in S(r)} c_y^i + \sum_{y \in \Lambda - S(r)} c_y^i\,, \tag{75}$$

for any $S(r) \equiv \{y \in \Lambda : |y - y_0| \le r\}$. For any $y \in \Lambda$, by definition of $c^i(s)$,

$$|\mathrm{tr}(\rho^s \widetilde{c}_y^i \rho^{1-s} O^\dagger O)| \le c^i(s)\,\mathrm{tr}(\rho\, O^\dagger O)\,. \tag{76}$$

The inequality (76) is good enough for the terms in $S(r)$. For the remaining terms with $y$ away from $y_0$ we have a better estimate because connected correlation decays when operators are far apart. There is a technical complication due to the fact that the factors of $\rho$ are separated by – and do not necessarily commute with – the $\widetilde{c}_y^i$. For this reason we need to use the Araki bound to show that operators remain sufficiently local under conjugation by the density matrix. Indeed by Lemma 2, $O^\dagger O \in \mathcal{B}(y_0, R; \xi, 4C)$ and from Theorem 4 and Corollary 1, there is $C_1(\mu_i, \xi, C, \varepsilon) > 0$ and $l(\mu_i) > 0$ such that for any $0 \le s \le 1$,

$$\|\rho^{-s} O^\dagger O \rho^s\| \le C_1 \|O^\dagger O\|(1 + 2R/a)\,, \tag{77}$$

$$\rho^{-s} O^\dagger O \rho^s \in \mathcal{B}\left(y_0, R + l(\mu_i) + a; \xi_A + \xi + \varepsilon, C_1 e^{\varepsilon R/(\xi_A + \xi)^2}\|O^\dagger O\|/\|\rho^{-s} O^\dagger O \rho^s\|\right)\,. \tag{78}$$

Hence by Theorem 5, because $\mathrm{tr}(\rho\, \widetilde{c}_y^i) = 0$, for any $0 \le s \le 1$,

$$\begin{aligned}
|\mathrm{tr}(\rho^s \widetilde{c}_y^i \rho^{1-s} O^\dagger O)| &= |\mathrm{tr}(\rho \rho^{-s} O^\dagger O \rho^s \widetilde{c}_y^i)| \\
&\le 2C_2 e^{R + l(\mu_i) + a + R_H + l_0(2) - |y - y_0|)/(\xi_{\mathrm{cor}} + \xi_A + \xi + \varepsilon)}\,,
\end{aligned} \tag{79}$$

where $C_2$ is defined in terms of the prefactor $C_{\mathrm{cor}}(\mu_i)$ in (38) as, using (77),

$$\begin{aligned}
C_2 &\equiv C_{\mathrm{cor}} \sup_{y \in \Lambda} \|\widetilde{c}_y^i\| \|\rho^{-s} O^\dagger O \rho^s\| + C_1 e^{\varepsilon R/(\xi_A + \xi)^2} \sup_{y \in \Lambda} \|\widetilde{c}_y^i\| \|O^\dagger O\| \\
&\le C_{\mathrm{cor}} C_1 \sup_{y \in \Lambda} \|\widetilde{c}_y^i\| \|O\|^2 (1 + 2R/a) + C_1 e^{\varepsilon R/(\xi_A + \xi)^2} \sup_{y \in \Lambda} \|\widetilde{c}_y^i\| \|O\|^2\,.
\end{aligned} \tag{80}$$

Now bound the sum (75) by choosing $r = R + l(\mu_i) + a + R_H + l_0(2) + \delta$ and apply (76) for $y \in S(r)$ and (79) for $y \notin S(r)$, (denote $\zeta \equiv \xi_{\mathrm{cor}} + \xi_A + \xi + \varepsilon$)

$$|\mathrm{tr}(\rho^s \widetilde{C}^i \rho^{1-s} O^\dagger O)| \le c^i(s)(1 + 2r/a)\mathrm{tr}(\rho\, O^\dagger O) + 4C_2 e^{-\delta/\zeta}(1 - e^{-a/\zeta})^{-1}\,, \tag{81}$$

and use the inequality (80) and $\|\widetilde{c}_y^i\| \le 2\|c_y^i\|$ to reduce to the form (70).

Proving (71) is essentially the same except in the first step:

$$\frac{\partial \mathcal{C}_{O_1 O_2}(\boldsymbol{x},t;\rho)}{\partial J_\alpha} = -\beta \int_0^1 ds\, \mathrm{tr}(\rho^s \widetilde{H}^\alpha \rho^{1-s} O^\dagger O) + 2\,\mathrm{Re}\,\mathrm{tr}\left(\rho\, O^\dagger \frac{\partial O}{\partial J_\alpha}\right)\,, \tag{82}$$

there is an additional term due to coupling dependence of $O_1(0, t)$. By definition,

$$\frac{\partial O}{\partial J_\alpha} = -\int_0^t ds \ [[H^\alpha(s), O_1(0, t)], O_2(\boldsymbol{x}, 0)] , \tag{83}$$

and (71) follows from the Cauchy-Schwartz inequality for the inner product $\langle O_1, O_2 \rangle \equiv \text{tr}(\rho \, O_1^\dagger O_2)$.

Finally if $C^i$ commute with each other, the first step (74) can be replaced with

$$\frac{\partial \mathcal{C}_{O_1 O_2}(\boldsymbol{x}, t; \rho)}{\partial \mu_i} = -\text{tr}(\sqrt{\rho} \widetilde{C}^i \sqrt{\rho} O^\dagger O) , \tag{84}$$

and the same proof goes through with $c^i$ as in (73). It is bounded by $2 \sup \|c_{\boldsymbol{y}}^i\|$ because $\sqrt{\rho} \, O^\dagger O \sqrt{\rho}$ is a positive operator and for any operator $S$ and positive operator $T$, $|\text{tr} ST| \leq \|S\| \text{tr} \, T$. $\qquad\square$

The theorem, as stated, seems complicated; but the physics is much clearer in terms of the velocity-dependent Lyapunov exponent (6):

**Corollary 2.** *For $v_S(\boldsymbol{v}; O_1, O_2, \xi)$ defined in (46),*

$$\left| \frac{\partial \lambda_{O_1 O_2}(\boldsymbol{v}; \rho)}{\partial \mu_i} \right| \leq \frac{2c^i}{a} \Big( v_S(\boldsymbol{v}; O_1, O_2, \xi) - \lambda_{O_1 O_2}(\boldsymbol{v}; \rho)(\xi_{\text{cor}} + \xi) \Big) . \tag{85}$$

*Proof.* Divide both sides of (70) by $t \, \mathcal{C}_{O_1 O_2}(\boldsymbol{x}, t; \rho)$, choose

$$\delta(t) = (\xi_{\text{cor}} + \xi_A + \xi + \varepsilon) \big[ -\lambda_{O_1 O_2}(\boldsymbol{v}; \rho) t + \varepsilon R/(\xi + \xi_A)^2 + \varepsilon t \big] > 0 , \tag{86}$$

$\boldsymbol{x} = \boldsymbol{v} t$ and take the limit $t \to \infty$ (assuming the limit and derivative commute):

$$\left| \frac{\partial \lambda_{O_1 O_2}}{\partial \mu_i} \right| \leq 2c^i \left\{ v_S + \big[ \varepsilon + \varepsilon v_S/(\xi + \xi_A)^2 - \lambda_{O_1 O_2} \big](\xi_{\text{cor}} + \xi_A + \xi + \varepsilon) \right\} /a . \tag{87}$$

Finally let $\varepsilon, \xi_A \to 0$ to conclude[5]. $\qquad\square$

The operator $O$ must decay at large distances at least as quickly as the rate set by $\xi_{\text{LR}}$ (appearing in any triple $(\boldsymbol{v}, \xi_{\text{LR}}, C_{\text{LR}})$ with a Lieb-Robinson bound Theorem 1). Therefore we take $\xi = \xi_{\text{LR}}$ in the main text. We have already noted in section C that this then defines a $v_S(\boldsymbol{v}; \xi) \leq v$.

The coupling dependence of $\lambda_{O_1 O_2}(\boldsymbol{v}; \rho)$ can be bounded in the same way:

**Corollary 3.** *If $\mathcal{C}_{O_1 O_2}(\boldsymbol{v} t, t; \rho) \sim \kappa_1^2 e^{\lambda_{O_1 O_2}(\boldsymbol{v}; \rho) t}$ and*

$$\mathcal{C}_{[H^\alpha(-s), O_1] O_2}(\boldsymbol{v} t, t; \rho) \sim \kappa_2^2 \|h^\alpha\|^2 e^{\lambda_{O_1 O_2}(\boldsymbol{v}; \rho) t} \tag{88}$$

*for $\kappa_1, \kappa_2 > 0$ and $\|h^\alpha\| \equiv \sup_{\boldsymbol{y} \in \Lambda} \|h_{\boldsymbol{y}}^\alpha\|$ at $t \to \infty$,*

$$\left| \frac{\partial \lambda_{O_1 O_2}(\boldsymbol{v}; \rho)}{\partial J_\alpha} \right| \leq \frac{2\beta h^\alpha}{a} \Big( v_S(\boldsymbol{v}; O_1, O_2, \xi) - \lambda_{O_1 O_2}(\boldsymbol{v}; \rho)(\xi_{\text{cor}} + \xi) \Big) + 2\|h^\alpha\| \kappa_2/\kappa_1 . \tag{89}$$

If we assume that the growth rate of the OTOC does not depend on choices of operators, i.e., the growth rate in (88) is $\lambda_{O_1 O_2}(\boldsymbol{v}; \rho)$, the same as that of $\mathcal{C}_{O_1 O_2}(\boldsymbol{v} t, t; \rho)$, this corollary shows that divergence of $\partial_J v_B$ at zero temperature pinpoints quantum phase transitions at which the system becomes gapless. Indeed, if to the contrary the system is gapped, as observed in Fig. 3 and discussed in the main text, the first term on the right side of (89) is expected to vanish at zero temperature so the right-hand side of (89) should be finite, contradicting the divergence of $\partial_J v_B$ via an inequality similar to (23). Cusps of scrambling characteristics are indeed observed at quantum critical points in e.g. [47, 48].

---

[5]Regarding the limit $\xi_A \to 0$ we refer readers to the discussions following Theorem 2.

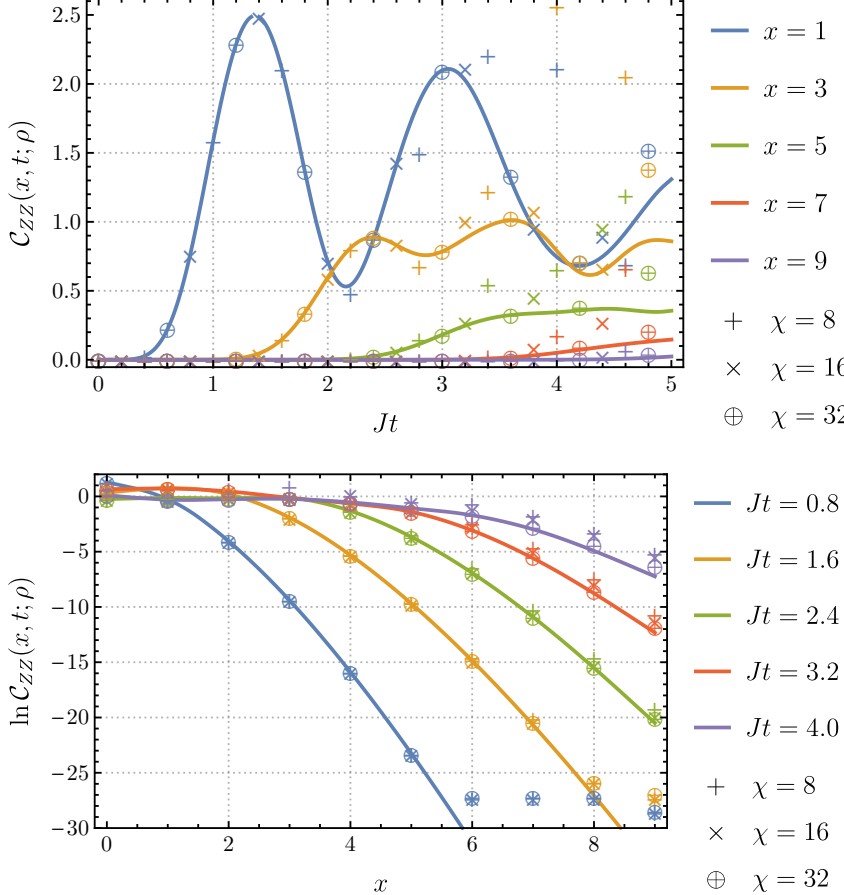

Figure 4: Comparison with Exact Diagonalization. The solid curves are from ED numerics in a mixed field Ising chain with $N = 10$, $h_X = 1.05J$ and $h_Z = 0.5J$ (see (21) for the Hamiltonian) and $\rho$ is the thermal state with $T = J$. In the first panel, each curve shows the time dependence of the OTOC at a fixed distance ($O_1 = Z_1$ and $O_2 = Z_{x+1}$). For finite bond dimension truncations $\chi = 8, 16$ and $32$, the MPO result agrees with ED at early times, and starts to deviate when the truncation is reached, which is near $Jt = 2, 3$ and $4$ respectively. In the second panel, each curve is a spatial profile of the OTOC at a fixed time. Propagation of a butterfly wavefront is clearly observed. For all $\chi$ the agreement with ED is remarkable until the MPO truncation $\varepsilon = 10^{-14}$ kicks in after $\ln \mathcal{C}$ drops to approximately $-25$.

# F  Numerical details

Our method is a generalization of the Matrix Product Operator (MPO) approach to calculating the butterfly velocity, presented in [19], to finite temperature states. The algorithm is implemented with the ITensor library, with operators $O_1(0, t)$, $O_2(\boldsymbol{x}, 0)$ and thermal density matrix $\rho$ represented as MPOs and evolved with a Time-Evolving Block Decimation (TEBD) method (for MPOs). For general quantum systems the thermal entanglement entropy is expected to be extensive. We find in practice that the MPO representation of thermal states works at sufficiently high but finite temperatures (in our case, $0 \leq \beta J \leq 3$). Numerical truncation $\varepsilon$ in the MPO is set to $\varepsilon = 10^{-14}$ and maximal bond dimension is denoted as $\chi = 256$. We will only investigate the mixed field Ising model with hopping $J$ and external fields $h_X$ and $h_Z$ as defined in (21), and probe the OTOC with Pauli $Z$ operators ($O_1 = O_2 = Z$ in (1)). Scrambling characteristics are then determined by least-squares fitting of $\ln \mathcal{C}$ at the wavefront to the

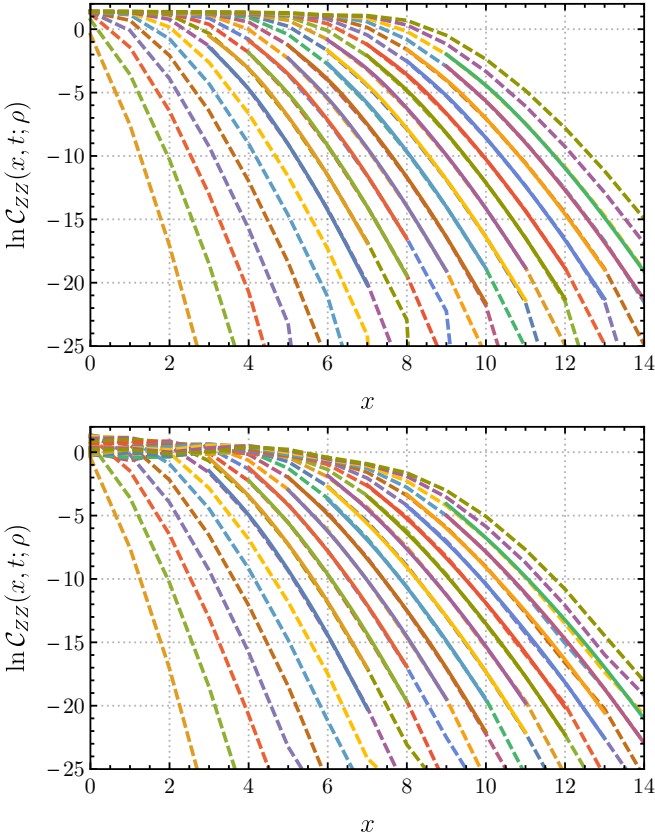

Figure 5: Examples of fitting. Dashed curves are from MPO numerics and fitting of (2) to wavefront is marked as solid. Each curve is $\ln \mathcal{C}$ for a fixed $Jt = 0.2, 0.4, \ldots, 4.8$. The first plot is for $\beta = 0$ and $h_X = 1.05J$, $h_Z = 0$ with a fitting $v_B = 1.95Ja$, $p = 0.46$ to be compared with exact values $v_B = 2Ja$ and $p = 0.5$ ($a = 1$ is the lattice spacing); the second plot is for $\beta J = 3$, $h_X = 1.05J$, $h_Z = 0.3J$ and the best fitting is $v_B = 1.39Ja$ with $p = 0.65$.

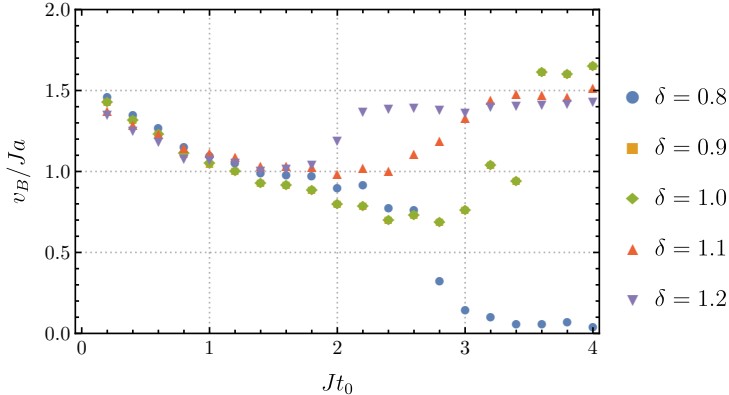

Figure 6: Fitted butterfly velocity at $h_X = 1.05J$, $h_Z = 0.4J$ and $\beta J = 3$ for different hyperparameters $\delta$ and $t_0$ ($Jt_1 = 4.4$ and $a = 1$). For small $t_0$, fluctuation with respect to $\delta$ is insignificant due to a larger amount of data. However, at these early times there is a systematic error leading to a dependence on $t_0$. When $Jt_0 > 2$ the fitting is not stable. The optimal choice of hyperparameters, from the figure, would be $Jt_0 \approx 1.5$ with $\delta \approx 1.0$.

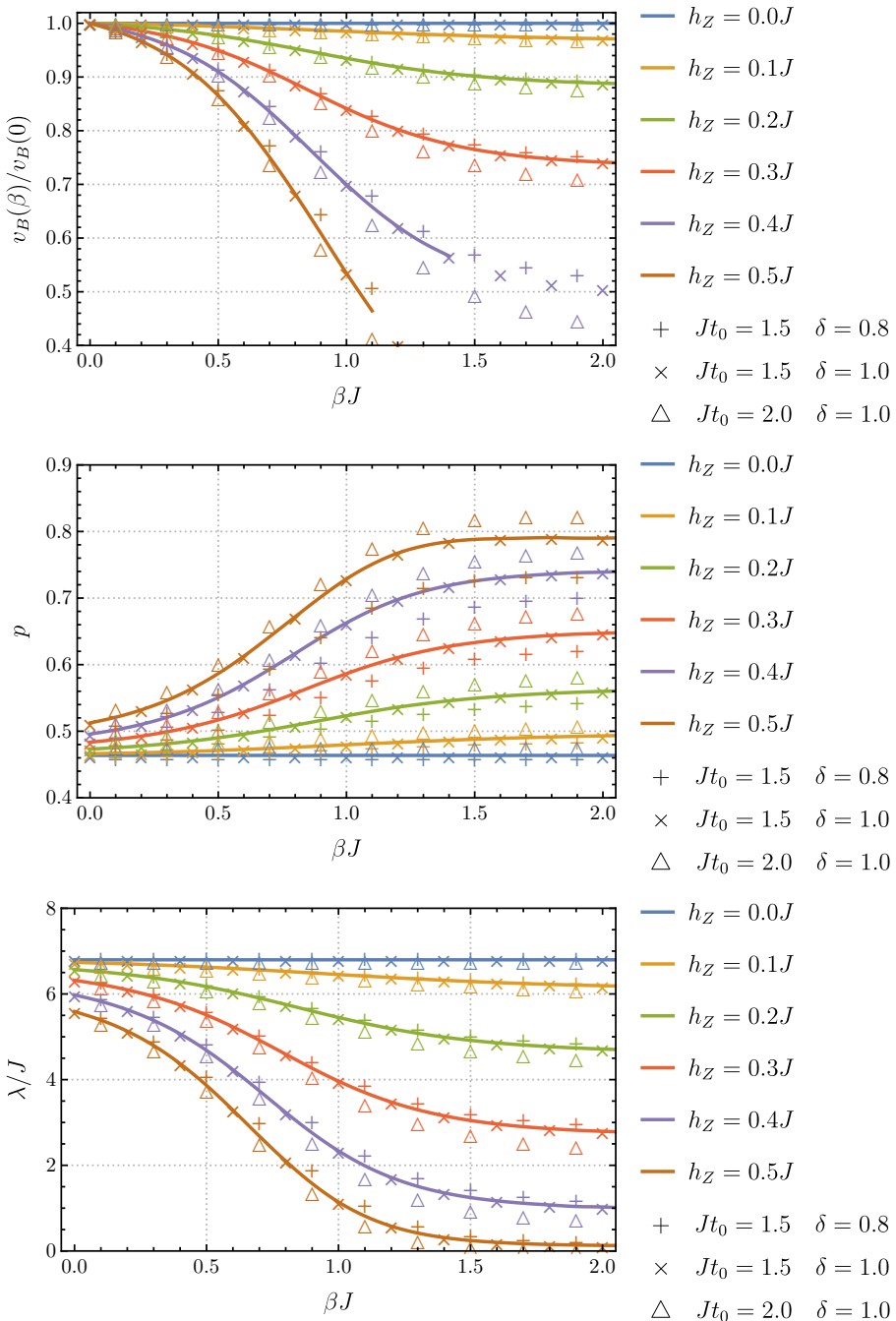

Figure 7: Scrambling characteristics in (2) fitted for numerics in mixed field Ising chain (21) with $h_X = 1.05J$, different longitudinal field $h_Z$, inverse temperature $\beta$ and hyperparameters $t_0$ and $\delta$ (with $Jt_1 = 4.4$). Solid curves are guides to the eye of fits at $Jt_0 = 1.5$ and $\delta = 1.0$.

expression (2).

The wavefront is determined as follows. First, due to numerical truncation with $\varepsilon = 10^{-14}$ only data with $\ln \mathcal{C} > -22$ will be used. This delimits the right end $r$ of the wavefront; the default left end $l_0$ is then defined as the position where $\partial_x \ln \mathcal{C}$ is half the value at $r$. To eliminate the arbitrariness of $l_0$ a hyperparameter $\delta > 0$ is introduced and the left end $l \equiv r - (r - l_0)\delta$. When $\delta = 1$, $l = l_0$ and when $\delta = 0$, $l = r$; hence $\delta$ tunes the range of the wavefront, ending

at $r$.

As a sanity check our implementation is verified against Exact Diagonalization (ED), which may be regarded as the MPO approximation with no bond dimension restrictions ($\chi = \infty$). The result is shown in Fig. 4. From the figure the MPO algorithm matches with ED perfectly at times before maximal bond dimension restriction is reached and starts to deviate afterwards. However, as shown in the figure, the wavefront dynamics is well captured by the MPO approximation, even after the bond dimension is saturated inside the butterfly cone. Such effectiveness of MPO (at least at infinite temperature) is observed in [19] and explained by the fact that at the wavefront the operator $O_1(0, t)$ is less complex, so only a smaller bond dimension is necessary.

A careful error analysis is necessary to extract reliable information from the nonlinear fit to the five parameters $(C, \lambda, x_0, v_B, p)$, appearing in (2). Here $C$ is the prefactor. Three major causes of systematic errors are identified: finite bond dimension $\chi$, a finite time range $[t_0, t_1]$ of data and inaccuracy of the functional form (2). The convergence with respect to bond dimensions is verified: for all data used the difference in $\ln \mathcal{C}$ between $\chi = 256$ and $\chi = 512$ is less than 0.05 and our main results do not depend on such a small difference. Also the fitting as presented in Fig. 5 is visually reasonably good, even for the chaotic Hamiltonian $h_Z = 0.3J$ at low temperature $\beta J = 3$.

The effect of a finite range of data and inaccuracy of the functional form is quantitatively manifested as dependence on the hyperparameters $\delta$ and $t_0$. Since the butterfly velocity is defined in the late time limit, $t_0$ should not be too small; but because only data up to time $t_1$ are available, $t_0$ cannot be arbitrarily large either. Moreover, larger $t_0$ means less data and more significant numerical instability. In Fig. 6, dependence on $\delta$ and $t_0$ of the fitted butterfly velocity for $\beta J = 3$ and $h_Z = 0.4J$ is shown. We will work with the values $\delta = 1.0$, $Jt_0 = 1.5$ and $Jt_1 = 4.4$.

With this choice of hyperparameters, we produce the figures in the main text. Errors are estimated via slightly tuning hyperparameters. Details are summarized in Fig. 7, with fitted values of $p$ and $\lambda$ given as well. From the plot errors are estimated to be within a scale of 0.05, 0.05 and 0.5 for $v_B(\beta)/v_B(0)$, $p$ and $\lambda/J$ respectively.

The correlation length $\xi$ is extracted with MPO numerics as well, as the inverse spatial decay rate of connected two-point correlations $\text{tr}(\rho Z_{15} Z_{15+x}) - \text{tr}(\rho Z_{15})\text{tr}(\rho Z_{15+x})$ in an $N = 50$ chain with operator insertions at sites 15 and $15 + x$, where $x = 0, 1, \ldots, 20$. The exponential fit is remarkably good with correlation lengths at different temperatures and longitudinal fields shown in Fig. 8. Given the correlation length $\xi$ along with $p$ and $\lambda$ from Fig. 7, the bound is evaluated (with error estimates) in Fig. 9. In evaluating the inequality (23) we have used $v_S \leq v$ for $v = 3Ja$ and $\xi_{\text{LR}} = a$ (cf. section C), where a Lieb-Robinson inequality with $(v, \xi_{\text{LR}}) = (3Ja, a)$ is verified in numerics and $a$ is the lattice spacing.

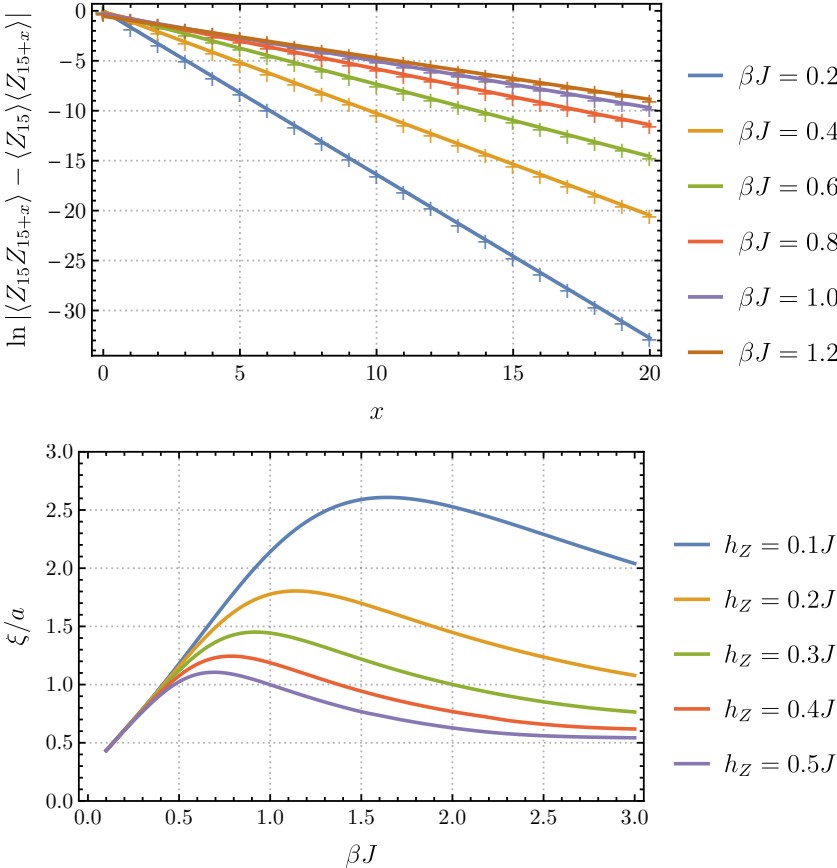

Figure 8: Lower plot: Correlation length $\xi$ for different inverse temperatures $\beta$ and longitudinal fields $h_Z$. $N = 50$, $h_X = 1.05J$ in (21) and $a$ is the lattice spacing. Upper plot: As an example, details of fitting at $h_Z = 0.1J$. + are numerical data and lines are linear fitting.

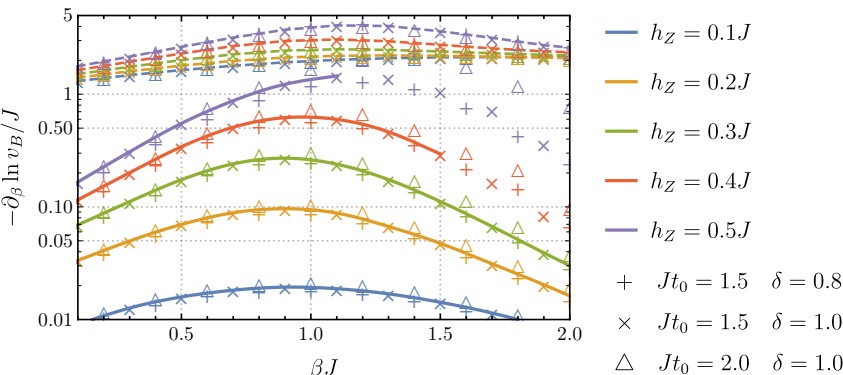

Figure 9: Temperature dependence of the butterfly velocity for different longitudinal fields $h_Z$ and hyperparameters $t_0$ and $\delta$ with $h_X = 1.05J$. Upper bounds are evaluated according to (23) shown as the dashed lines in the top of the figure.

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
