# Peer review of "Quantum Scrambling and State Dependence of the Butterfly Velocity"

_SciPost Physics, doi:SciPost Phys. 7, 045 (2019)_

## Round 2 · Referee Report · Anonymous (Referee 1) · 2019-5-27

Strengths

See report

Weaknesses

See report

Report

Many-body quantum dynamics and operator spreading are topics of much current interest. Most analytic treatments of these subjects appeal to various tractable limits in special models (such as large N and low temperature in the SYK model). Complementary approaches using "random-circuit" models have been used to make predictions about "strongly quantum" many-body systems away from large N/weak coupling limits, but these models are often random in time so that energy is not conserved and hence one must work with the equal-weight infinite temperature ensemble. As such, analytic results on strongly quantum systems at finite temperature are sorely lacking.

The paper by Han and Hartnoll attempts to address this gap by studying the temperature dependence of the butterfly velocity. They introduce a new quantity, the scrambling velocity, and use this to prove that the rate of change of the butterfly speed with temperature is upper bounded by the scrambling velocity. In other words, models with a zero scrambling velocity should have a temperature independent butterfly speed. This is an interesting result that is analytically derived and numerically verified, and I recommend publication. However, the paper would be strengthened if the following points could be addressed:

  1. While the notion of the scrambling velocity and its relation to the state dependence of the butterfly speed is conceptually interesting, it would be useful to have some concrete examples of systems where $v_S$ is actually zero (besides the free-field example mentioned in the paper). While the introduction to the paper seems to be motivated by the question of scrambling in strongly quantum lattice models, it seems as though all such lattice models have non-zero scrambling velocity (?) As the authors discuss, even non-interacting systems with a quasiparticle description can have a non-zero $v_S$, and the temperature independence of $v_B$ in these systems has to then be separately argued for.

  2. Following up on 1, do the authors see a way to generalize their result so that the free case directly follows instead of needing extra discussion? For example, perhaps the bound can be made for the slowest scrambling speed over the space of all operators? I'm also confused by real vs. momentum space. The OTOC is defined for local operators in real space, while quasiparticles of free theories generally live in momentum space. Won't it be the case that acting with a local operator creates quasiparticles of all momenta and hence gives a non-zero scrambling speed? I understand the semi-classical intuition that a quasiparticle with a (q,x) propagates, but does not grow, in space so that free systems should intuitively have $v_S=0$. However, I'm not sure if the definition for $v_S$ in (8) actually captures this.

  3. The LHS of equation (9), $|\partial_\beta \lambda(v;\rho)|$, is positive. The RHS is $\frac{2h}{a}(v_s(v) - (\xi +\xi_{LR})\lambda(v)).$ Should there be an absolute value surrounding $\lambda(v)$ on the RHS? $\lambda(v)$ can be negative, in which case $|\partial_\beta \lambda(v;\rho)|$ wouldn't be upper bounded by $v_S$.

  4. The authors use the words "quantum chaos" quite casually in the introduction, using it almost synonymously with scrambling or the growth of the OTOC. However, interacting integrable systems also scramble operators and show a growth in the OTOC, but are non-chaotic. Indeed, while there are well-defined measures of intermediate-to-late time chaos in MB systems (say as diagnosed by the spectral form factor for times greater than the Thouless time, or by nearest neighbor eigenenergy level repulsion), the connections of these measures of chaos to the early-time growth of the OTOC has not been conclusively established. It would be helpful for the authors to separate "scrambling" from "chaos" in their discussions.

  5. On the last paragraph of page 2 which summarizes results, "velocity dependent Lyapunov exponent" is used before it is defined.

Requested changes

See report

---

## Round 2 · Referee Report · Anonymous (Referee 2) · 2019-6-12

Strengths

See report

Weaknesses

See report

Report

The growth of out-of-time-order correlators has been shown to be intimately connected to the scrambling of quantum information, whereby the information in the local degrees of freedom of a quantum many-body system is spread over its many degrees of freedom. These concepts are of great interest due to the connection between the fast-scramblers and black-holes, as well as the intimate connection between scrambling and the dynamics of entanglement entropy and thermalization. So far progress has been made on two separate fronts: in the large N or weak-coupling regimes, or a regime no immediately relevant to the original scrambling bounds, probing scrambling in quantum simulators starting with pure states.
Here Han and Hartnoll study the temperature dependence of the butterfly velocity in finite systems in 1D (both numerical and analytical results). I found this discussion quite interesting, and relatively novel. Their proof relies on a new quantity called the scrambling velocity which allows them to bound the rate of change of the butterfly velocity as a function of the temperature.

Requested changes

I think the paper is suitable for publication however I would like the authors to address the following points:

  1. I find the motivation for the scrambling velocity confusing. If $v_s$=0 then $v_B$ is temperature independent. The authors state that this is often true in non-interacting systems. However, I think this is only true for arbitrary state and operators, only in non-interacting systems. Is that correct? Since any unitary dynamics which generates entanglement should in principle lead to non-zero $v_s$.
  2. In the case of non-interacting models where $v_s$ is non-zero but $v_B$ is independent of temperature can the authors provide some more intuition? I understand the quasi-particle picture however are there classes of spin models the authors suspect may present such characteristics?
  3. The authors at the start of the paper focused on chaotic systems. Can they specifically define what they mean by this? That is, do they require exponential growth of the OTOC or saturating some bound for the Lyapunov exponent? later on they study an integrable system which is not chaotic so I was slightly confused by the initial focus on the chaotic systems.
  4. Finally how important is the notion of temperature in the results? In many quantum simulation platforms the preparation of a pure state at a specific energy is accessible. Can the results be reformulated in terms of energy of the pure state?

---

## Round 3 · Author Response

We thank the referees for their constructive comments. We have made fairly extensive changes to the presentation of the paper to address the issues raised. We believe that the paper has been improved. The technical content is unchanged.
A. Both referees commented that we were using the words “quantum chaos” quite loosely. We agree with this comment. We have removed all references to chaos from the introduction and focused on notions of scrambling as defined by the OTOC.
B. Both referees commented that the conditions for vS = 0 were unclear, possibly undermining the usefulness of the scrambling velocity concept that we have introduced. This is a fair comment. In terms of the usefulness of our bound (on the temperature dependence of scrambling, in terms of vS), we mostly had in mind strongly scrambling systems. It is correct to say that for weakly scrambling systems, and even non-interacting systems, vS as currently defined may not be small, and therefore the bound is less useful. We have restructured the text to emphasize the strong scrambling case, and the connection to the numerics we performed, and to clarify the limitations of our current definition away from that limit (see for example the final paragraph on the section defining the scrambling velocity). We have also modified the discussion of the non-interacting model (discussion below fig 2) to make clearer why vS = 0 in that case.
Finally there were the following more minor comments from the referees:
C. One of the referees asked whether our results can be reformulated in terms of the energy of a pure state rather than the temperature. We do not have anything to contribute here beyond the usual expectation that the canonical and microcanonical ensembles should be equivalent for local observables in thermalizing systems.
D. One of the referees noted that "velocity dependent Lyapunov exponent" was used before it was defined. We have added a reference to the definition later in the text.
E. One of the referees asked whether there should be absolute value surrounding lambda on the RHS of equation (9), in order to obtain an upper bound in terms of vS. In general both terms on the RHS of equation (9) are important. We show in the text — around equation (23) — that in zooming in to the butterfly light cone, it is the first term that becomes important. This is why the bound is in terms of vS. The sign of lambda doesn’t matter here.
A. Both referees commented that we were using the words “quantum chaos” quite loosely. We agree with this comment. We have removed all references to chaos from the introduction and focused on notions of scrambling as defined by the OTOC.
B. Both referees commented that the conditions for vS = 0 were unclear, possibly undermining the usefulness of the scrambling velocity concept that we have introduced. This is a fair comment. In terms of the usefulness of our bound (on the temperature dependence of scrambling, in terms of vS), we mostly had in mind strongly scrambling systems. It is correct to say that for weakly scrambling systems, and even non-interacting systems, vS as currently defined may not be small, and therefore the bound is less useful. We have restructured the text to emphasize the strong scrambling case, and the connection to the numerics we performed, and to clarify the limitations of our current definition away from that limit (see for example the final paragraph on the section defining the scrambling velocity). We have also modified the discussion of the non-interacting model (discussion below fig 2) to make clearer why vS = 0 in that case.
Finally there were the following more minor comments from the referees:
C. One of the referees asked whether our results can be reformulated in terms of the energy of a pure state rather than the temperature. We do not have anything to contribute here beyond the usual expectation that the canonical and microcanonical ensembles should be equivalent for local observables in thermalizing systems.
D. One of the referees noted that "velocity dependent Lyapunov exponent" was used before it was defined. We have added a reference to the definition later in the text.
E. One of the referees asked whether there should be absolute value surrounding lambda on the RHS of equation (9), in order to obtain an upper bound in terms of vS. In general both terms on the RHS of equation (9) are important. We show in the text — around equation (23) — that in zooming in to the butterfly light cone, it is the first term that becomes important. This is why the bound is in terms of vS. The sign of lambda doesn’t matter here.

---

## Round 3 · List of Changes

See reply above.

---

## Editorial Decision

published